# Disentangling Latent Shifts of In-Context Learning with Weak Supervision

**Josip Jukić, Jan Šnajder**
TakeLab
Faculty of Electrical Engineering and Computing
University of Zagreb, Croatia
{josip.jukic, jan.snajder}@fer.hr

## Abstract

In-context learning (ICL) enables large language models to perform few-shot learning by conditioning on labeled examples in the prompt. Despite its flexibility, ICL suffers from instability – especially as prompt length increases with more demonstrations. To address this, we treat ICL as a source of weak supervision and propose a parameter-efficient method that disentangles demonstration-induced latent shifts from those of the query. An ICL-based teacher generates pseudo-labels on unlabeled queries, while a student predicts them using only the query input, updating a lightweight adapter. This captures demonstration effects in a compact, reusable form, enabling efficient inference while remaining composable with new demonstrations. Although trained on noisy teacher outputs, the student often outperforms its teacher through pseudo-label correction and coverage expansion, consistent with the weak-to-strong generalization effect. Empirically, our method improves generalization, stability, and efficiency across both in-domain and out-of-domain tasks, surpassing standard ICL and prior disentanglement methods.

## 1 Introduction

In-context learning (ICL) has become a core mechanism for adapting large language models (LLMs) to new tasks in a way that removes the need to update their parameters [3, 10]. By prepending a few labeled examples, called *demonstrations*, to the input query, LLMs can perform few-shot learning directly at inference time. This paradigm is especially attractive in low-resource settings, where full fine-tuning is too costly or impractical.

Despite its convenience, ICL performance is highly sensitive to the selection and ordering of demonstrations, often resulting in unstable predictions and poor generalization [30, 25]. Moreover, ICL typically requires long contexts, as multiple demonstrations must be included alongside the query in a single input. As input lengths grow, inference becomes increasingly inefficient, inflating processing costs, amplifying positional biases – including inherent primacy and recency effects in transformer-based LLMs [27] – and pushing against the model's context window limits [11]. Consequently, ICL scales poorly with the number of demonstrations [5, 4]: beyond a certain threshold, additional examples either degrade performance or must be discarded entirely. This inefficiency and poor scalability limit ICL's ability to incorporate more supervision, preventing it from fully leveraging the potential benefits of richer demonstrations.

To address these limitations, a mechanistic perspective on ICL has proven useful. In this view, demonstrations influence model behavior by inducing *latent shifts* – context-dependent changes in internal representations that alter how the model processes the query. Disentangling these shifts from the representation of the query itself allows ICL to operate more robustly, processing queries independently of demonstrations. This, in turn, enables contextual knowledge to be stored persistently,

39th Conference on Neural Information Processing Systems (NeurIPS 2025).

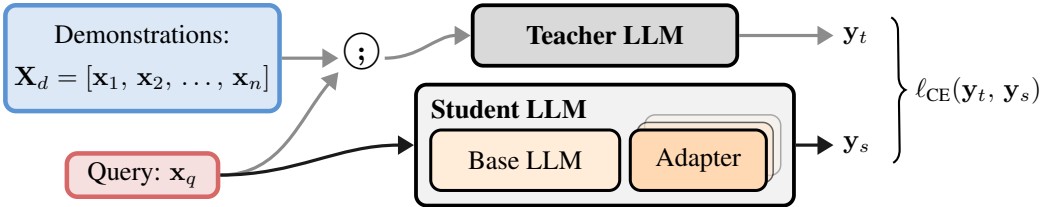

Figure 1: Illustration of WILDA. The teacher processes a concatenation (denoted by ⨾) of demonstrations $\mathbf{X}_d$, consisting of $n$ demonstrations $[\mathbf{x}_1, \mathbf{x}_2, \ldots, \mathbf{x}_n]$, and the query $\mathbf{x}_q$. The student, using only the query, fine-tunes its adapter weights to produce outputs $\mathbf{y}_s$ aligned with the teacher's pseudo-labels $\mathbf{y}_t$ by minimizing the cross-entropy loss $\ell_{\mathrm{CE}}$.

eliminating the need to reprocess demonstrations for each new query. The latent shifts can then be reapplied directly, thereby reducing prompt length, improving inference efficiency, and enabling more modular and reusable representations. The disentanglement of demonstration-induced latent shifts from those of the query has been explored from both practical and theoretical perspectives: some work aims to improve the stability and scalability of ICL [28, 42], while other studies provide formal insights into the nature of context-induced shifts [8, 34]. However, existing methods rely on approximations that intervene directly in attention heads or hidden states. In contrast, we adopt a functional perspective on ICL, capturing demonstration-induced shifts implicitly through model outputs rather than architectural manipulation. This enables disentanglement to emerge as a learned, self-aligned process rather than one achieved through explicit manipulation of internal states.

In this paper, we propose to disentangle the latent shifts of the demonstrations and the query by using the output of ICL as a weak supervision signal, rather than approximating the underlying internal mechanisms. Specifically, we use ICL predictions as pseudo-labels that capture the full, contextualized influence of demonstrations. These pseudo-labels guide the training of a student model that internalizes demonstration-conditioned behavior – without repeated prompting or architectural intervention. We instantiate this idea with WILDA (**W**eakly-supervised **I**n-context **L**earning **D**isentanglement via **A**dapters), a parameter-efficient method for encoding the latent shifts induced by in-context demonstrations. In a teacher–student setup, an ICL-based teacher generates pseudo-labels for unlabeled queries, while the student learns to predict them using only the query input. The student updates a lightweight adapter module [16], enabling it to capture the shift in a reusable and modular form. The adapter supports efficient inference without requiring demonstrations in the prompt, and multiple adapters can be combined through simple arithmetic to fuse knowledge across demonstration subsets. At inference time, additional in-context demonstrations can be composed with the adapter's shift, enabling flexible integration of prompt-based and parameter-based adaptation.

We evaluate WILDA on both in-domain (ID) and out-of-domain (OOD) data, comparing it to standard ICL, prompt-based fine-tuning [33], and recent approaches that manipulate architecture or hidden states to disentangle latent shifts. WILDA consistently improves generalization, prompt robustness, and inference efficiency, while remaining highly parameter-efficient. Despite learning from ICL outputs, the student often surpasses its teacher. We experimentally show that this improvement arises from two emergent behaviors: *pseudo-label correction*, where the student refines noisy or inconsistent outputs from the teacher, and *coverage expansion*, where it generalizes beyond the narrow patterns encoded in the demonstrations. Together, these effects enable *weak-to-strong (W2S) generalization* [23], allowing the model to learn stable task behavior from limited supervision.

Our contribution is threefold: (1) we propose WILDA, a method that encodes ICL-induced behavior into reusable adapters, improving inference efficiency and prompt stability without requiring demonstration prompts; (2) we show that WILDA outperforms traditional ICL and latent shift disentanglement methods on both ID and OOD data, while maintaining parameter efficiency; and (3) we demonstrate that multiple adapters can be combined through simple arithmetic operations, enabling scalable composition of demonstration subsets and efficient handling of long-context scenarios. Together, these results demonstrate that viewing ICL as a form of weak supervision enables parametric generalization that can also flexibly incorporate contextual adaptation. WILDA thus offers a scalable and composable framework for stable task adaptation in LLMs.[1]

---

[1]Our code is available at `https://github.com/josipjukic/wilda`.

## 2 Method

### 2.1 Disentangling Latent Shifts

Disentangling in-context knowledge from the query can enhance the efficiency and stability of ICL. Current methods typically achieve disentanglement by modifying the outputs of attention heads or hidden states. The theoretical motivation for this approach stems from previous studies [2, 20], which demonstrate that linear layers optimized via gradient descent can be viewed through the lens of linear attention mechanisms. Specifically, consider a neural network's linear layer characterized by an initial weight matrix $\mathbf{W}_0 \in \mathbb{R}^{m \times n}$ and an update $\Delta \mathbf{W} \in \mathbb{R}^{m \times n}$ resulting from backpropagation. Given an input representation $\mathbf{x} \in \mathbb{R}^m$, the linear transformation $\mathbf{f} : \mathbb{R}^m \to \mathbb{R}^n$ can be expressed succinctly as $\mathbf{f}(\mathbf{x}) = (\mathbf{W}_0 + \Delta \mathbf{W})\mathbf{x}$. Let $\mathbf{x}_i \in \mathbb{R}^m$ be a training example and $\mathbf{e}_i \in \mathbb{R}^n$ the error signal on $\mathbf{x}_i$ obtained from the gradient of the loss function. During backpropagation, $\Delta \mathbf{W}$ is computed by accumulating the outer products (denoted by $\otimes$) of $N$ training examples $\{\mathbf{x}_1, \mathbf{x}_2, \dots, \mathbf{x}_N\}$ and their error signals $\{\mathbf{e}_1, \mathbf{e}_2, \dots, \mathbf{e}_N\}$, i.e., $\Delta \mathbf{W} = \sum_{i=1}^{N} \mathbf{e}_i \otimes \mathbf{x}_i$. The update part of linear layers optimized by gradient descent can be expressed as unnormalized linear dot-product attention [20]:

$$\mathbf{f}(\mathbf{x}) = (\mathbf{W}_0 + \Delta \mathbf{W})\mathbf{x} = \mathbf{W}_0 \mathbf{x} + \sum_{i=1}^{N} (\mathbf{e}_i \otimes \mathbf{x}_i)\mathbf{x} = \mathbf{W}_0 \mathbf{x} + \underbrace{\sum_{i=1}^{N} \mathbf{e}_i (\mathbf{x}_i^T \mathbf{x})}_{\text{linear attention}} . \quad (1)$$

In the context of the attention mechanism, this shows that the latent shift $\Delta \mathbf{W} \mathbf{x}$ induced by training examples corresponds directly to the application of linear attention, with error signals $\mathbf{e}_i$ as values, training examples $\mathbf{x}_i$ as keys, and the current input $\mathbf{x}$ as the attention query.

The concept of disentangling the latent shifts described in (1) can be extended to ICL, albeit only under the approximation of linear attention. Let $\mathbf{W}_V$, $\mathbf{W}_K$, and $\mathbf{W}_Q$ denote the weight matrices for values, keys, and queries, respectively. Let $\mathbf{x}_q^{(t)}$ represent the current query token's embedding at step $t$, and $\mathbf{q}^{(t)} = \mathbf{W}_Q \mathbf{x}_q^{(t)}$ is the corresponding attention query vector. The matrix $\mathbf{X}_q = [\mathbf{x}_q^{(1)}, \mathbf{x}_q^{(2)}, \dots, \mathbf{x}_q^{(t-1)}]$ contains all previous query token representations up to $t-1$, and $\mathbf{X}_d$ is the matrix of demonstration token representations. The concatenation $[\mathbf{X}_d; \mathbf{X}_q]$ along the sequence dimension is used to compute the output of a single attention head (AH) at step $t$, expressed as:

$$\mathbf{f}_{\text{AH}}(\mathbf{x}_q^{(t)}) = \mathbf{W}_V [\mathbf{X}_d; \mathbf{X}_q] \, \text{softmax}\left( \frac{(\mathbf{W}_K [\mathbf{X}_d; \mathbf{X}_q])^\top \mathbf{q}^{(t)}}{\sqrt{d}} \right), \quad (2)$$

where $d$ is the scaling factor (i.e., the dimensionality of the key vectors). By approximating the attention mechanism with linear attention, it becomes possible to disentangle the latent shift of the zero-shot output of an attention head induced by the query from the latent shift induced by the demonstrations [8]:

$$\begin{aligned}
\mathbf{f}_{\text{AH}}(\mathbf{x}_q^{(t)}) &\approx \mathbf{W}_V [\mathbf{X}_d; \mathbf{X}_q] \left( \mathbf{W}_K [\mathbf{X}_d; \mathbf{X}_q] \right)^\top \mathbf{q}^{(t)} \\
&= \underbrace{\mathbf{W}_V \mathbf{X}_q \left( \mathbf{W}_K \mathbf{X}_q \right)^\top}_{\mathbf{W}_{\text{ZS}}} \mathbf{q}^{(t)} + \underbrace{\mathbf{W}_V \mathbf{X}_d \left( \mathbf{W}_K \mathbf{X}_d \right)^\top}_{\Delta \mathbf{W}_{\text{ICL}}} \mathbf{q}^{(t)} .
\end{aligned} \quad (3)$$

This approximation disentangles the latent shift induced by the demonstrations $\mathbf{X}_d$ from that induced by the query $\mathbf{x}_q^{(t)}$ (see Appendix A for a detailed derivation of (3)). *The contribution from ICL is captured as a virtual weight update $\Delta \mathbf{W}_{\text{ICL}}$, corresponding to virtual gradients*, often referred to as "meta-gradients" in the literature. The zero-shot latent shift of the query, corresponding to $\mathbf{W}_{\text{ZS}} \mathbf{q}^{(t)}$, reflects the output without demonstrations, providing the initial state. Analogous to $\Delta \mathbf{W} \mathbf{x}$ in (1), the latent shift $\Delta \mathbf{W}_{\text{ICL}} \mathbf{q}^{(t)}$ reflects the contribution of ICL. Finally, by substituting $\mathbf{h}_{\text{ZS}} = \mathbf{W}_{\text{ZS}} \mathbf{q}^{(t)}$ and $\Delta \mathbf{h}_{\text{ICL}} = \Delta \mathbf{W}_{\text{ICL}} \mathbf{q}^{(t)}$, we can rewrite the output of an attention head as:

$$\mathbf{f}_{\text{AH}}(\mathbf{x}_q^{(t)}) \approx \mathbf{h}_{\text{ZS}} + \Delta \mathbf{h}_{\text{ICL}} . \quad (4)$$

Although transformer-based LLMs employ non-linear attention in practice, many methods [8, 42, 34] rely on theoretical assumptions from linear attention, manipulating attention heads or hidden states to approximate latent shift disentanglement. This simplification, however, overlooks key architectural

components such as feed-forward layers, activation functions, and residual connections. While effective to a degree, these methods fall short of fully capturing the complex dynamics through which transformers process demonstrations. In this work, we explore how virtual weight updates can be obtained more directly while preserving the key components of the transformer architecture.

## 2.2 Weak Supervision with ICL

To disentangle the latent shifts induced by in-context demonstrations, we introduce WILDA, a method that uses ICL predictions as a form of weak supervision to encode these shifts into reusable adapter parameters [16, 17]. Instead of focusing narrowly on attention head manipulations, WILDA captures the full impact of demonstrations as expressed in the model's final outputs – reflecting the combined effects of all components, including attention layers, feed-forward blocks, and residual paths. By aligning with the actual latent shifts induced by ICL, WILDA enables the model to embed and reapply in-context knowledge using its full architecture, without relying on repeated prompting.

At the core of WILDA is a simple teacher–student framework: the teacher model, $\mathbf{f}_{\text{teacher}}$, processes both the demonstrations and the query together to generate pseudo-labels without requiring additional labeled data. The student model, $\mathbf{f}_{\text{student}}$, shares the same architecture as the teacher but includes adapter parameters. Unlike the teacher, the student processes only the query, using the adapter to internalize the knowledge from the demonstrations, as illustrated in Figure 1. Let $\mathbf{x}_q$ denote the query input and $\mathbf{X}_d$ the matrix of demonstration tokens, where each row corresponds to a single demonstration. The empirical loss is defined using the cross-entropy loss function $\ell_{\text{CE}}$, which aligns the student's output distribution with the teacher's full probability distribution over the vocabulary. This enables the student to learn from the full signal provided by the teacher's output logits. Formally, the empirical loss is

$$\sum_{\mathbf{x}_q \in \mathcal{D}_{\text{unlab}}} \ell_{\text{CE}}\left(\mathbf{f}_{\text{teacher}}\left([\mathbf{X}_d^*; \mathbf{x}_q]\right), \mathbf{f}_{\text{student}}\left(\mathbf{x}_q\right)\right), \tag{5}$$

where $\mathcal{D}_{\text{unlab}}$ is an unlabeled dataset and $\mathbf{X}_d^*$ is a flattened version of $\mathbf{X}_d$.

WILDA fundamentally differs from existing approaches, which manipulate attention heads or hidden states at query time, by instead progressively embedding the knowledge from demonstrations into the adapter parameters, denoted $\mathbf{W}_{\text{ICL}}$. The base LLM parameters, $\mathbf{W}_{\text{ZS}}$, capture the zero-shot component, while the total model parameters may be represented as $\mathbf{W}_{\text{ZS}} \oplus \mathbf{W}_{\text{ICL}}$, where $\oplus$ denotes the composition of base and adapter parameters.[2] This setup captures the latent shift introduced by the demonstrations through $\mathbf{W}_{\text{ICL}}$, extending the disentangling process outlined by (3) across the model's entire architecture. The teacher processes the full input sequence $[\mathbf{X}_d^*; \mathbf{x}_q]$, while the student processes only the query, applying $\mathbf{W}_{\text{ICL}}$ to integrate demonstration knowledge without explicitly processing the demonstrations. Analogously to (4), the latent shift induced by demonstrations can be recovered by decomposing outputs into zero-shot and ICL components. Let $\mathbf{h}_{\text{LLM}}(\mathbf{x}_q \mid \mathbf{W})$ represent the final latent states of an LLM with parameters $\mathbf{W}$ when processing the input $\mathbf{x}_q$. The following decomposition holds:

$$\mathbf{h}_{\text{LLM}}(\mathbf{x}_q \mid \mathbf{W}_{\text{ZS}} \oplus \mathbf{W}_{\text{ICL}}) = \mathbf{h}_{\text{LLM}}(\mathbf{x}_q \mid \mathbf{W}_{\text{ZS}}) + \Delta\mathbf{h}_{\text{ICL}}, \tag{6}$$

where $\Delta\mathbf{h}_{\text{ICL}}$ encapsulates the latent shift attributable to the demonstrations. WILDA encodes the latent shift implicitly within the adapter parameters $\mathbf{W}_{\text{ICL}}$, which is central to our approach. However, if necessary, the latent shift can also be explicitly calculated owing to the decomposition in (6).

WILDA achieves stability not only by disentangling demonstration effects, but also through its training dynamics and parametric nature. During training, the same LLM instance serves as both teacher and student across epochs, with the adapter toggled on or off to alternate between roles. Shuffling demonstrations between epochs mitigates order sensitivity, further stabilizing the ICL process. Crucially, WILDA leverages its parametric adapter to internalize demonstration-induced shifts, enabling the model to generalize effectively across ID and near-OOD data (see Section 3). This aligns naturally with the W2S generalization paradigm [23], in which the student is not merely expected to match the teacher but to surpass it. WILDA facilitates this process by compactly encoding latent shifts in a way that supports both pseudo-label correction (refining noisy targets through local consistency) and coverage expansion (generalizing beyond the teacher's original scope). Together, these effects enable stable extrapolation across the data distribution, aligning with theoretical expectations of W2S generalization [39].

---

[2]Notably, the number of adapter parameters is significantly smaller compared to the base model parameters.

# 3 Experiments

## 3.1 Experimental Setup

We perform our experiments using decoder-only autoregressive language models provided by Hugging Face [41]. Specifically, we employ Llama 3 (8B) [12] and Phi 3 (mini 4k) [1], along with Llama 2 (7B) [35] for comparative purposes. Further details about the models are listed in Table 13 of the Appendix.

We assess model performance on seven tasks from the GLUE benchmark [37], covering single-sequence binary classification (COLA, SST, RTE), sequence-pair binary classification (MRPC, QQP, QNLI), and sequence-pair multi-class classification (MNLI). Evaluation metrics follow established standards: Matthew's correlation for COLA, $F_1$ scores for MRPC and QQP, and accuracy for the remaining tasks, with evaluations conducted on the development sets. Additionally, we measure accuracy on selected datasets from the MMLU benchmark [14], specifically "elementary math" (MATH) and "miscellaneous" (MISC). We further extend our analysis to the ARC-Challenge benchmark [7] to assess reasoning and multi-hop generalization, which we show in Appendix D.

Predictions are made based on the probability of generating specific verbalizer tokens as the first token output by the models, facilitated by carefully crafted prompts designed explicitly for single-token answers (see Appendix F for detailed templates).

Our experiments compare WILDA with several baselines, including **Zero-Shot (0-shot)** inference, which generates predictions without demonstrations, **Standard ICL (n-shot)**, which uses $n$ demonstrations at inference time, and **Pattern-Based Fine-Tuning (PBFT)** [33], which fine-tunes an adapter module on data-specific patterns. We also include two ICL disentanglement methods, **In-Context Vectors (ICV)** [28], which leverages hidden-state representations from demonstration examples, and **Batch-ICL** [42], which aggregates meta-gradients across multiple one-shot runs. All methods are evaluated using a fixed number of demonstrations, with $n \in \{4, 8, 16, 32\}$. Each experiment is repeated 10 times with different random seeds, resulting in varied demonstration selections across runs. Alongside generalization scores, we report the standard deviation across these runs as an indicator of each method's stability. Evaluations for GLUE are conducted on the development sets, whereas for the MMLU datasets, we randomly sample 200 instances for evaluation.

We employ three variants of WILDA, each differing in how demonstrations are selected or ordered during training: **fixed (WILDA-F)** uses a fixed, unchanging set of demonstrations throughout the entire training process, **shuffle (WILDA-S)** uses the same demonstrations throughout training but shuffles their order at the beginning of each epoch, and **resample (WILDA-R)** draws a new set of demonstrations from a larger labeled pool at each epoch.

We utilize LoRA (Low-Rank Adaptation) [17] for the adapter modules (for both PBFT and WILDA), corresponding to 0.1–0.3% of the total parameter count, depending on the model (see Table 13 in the Appendix for adapter sizes per model). For each task, we generate pseudo-labels using the teacher model on unlabeled data. Specifically, we use 100 unlabeled instances ($\mathcal{D}_{\text{unlab}}$ in (5)) for both the GLUE and MMLU benchmarks. Additionally, for GLUE datasets, we experiment with 200 and 500 instances to assess the impact of the amount of unlabeled data on generalization and stability. We experiment only with 100 unlabeled instances for MMLU datasets due to their limited size. In all of the experiments, we fine-tune the adapter for 10 epochs. Further experimental details are provided in Appendix E.

## 3.2 Generalization and Stability

We first evaluate the generalization and stability of WILDA on ID data. Table 1 reports the 16-shot ID generalization scores along with standard deviations. Across all datasets and models, WILDA-S consistently achieves the best generalization scores, outperforming standard ICL, PBFT, and the disentanglement methods ICV and Batch-ICL (cf. Table 6 in the Appendix for results with Llama 2). Compared to standard ICL, WILDA-S *shows absolute improvements ranging from* 2.6% *to* 11.9% *for Llama 3 and* 2.5% *to* 10.3% *for Phi 3*, where the differences in scores are statistically significant across all datasets.[3] Similar patterns hold for $n \in \{4, 8, 32\}$, where WILDA-S also surpasses standard

---

[3]We assess the statistical significance using a two-tailed Wilcoxon signed-rank test ($p < 0.05$), applying the Holm-Bonferroni method for family-wise error rate correction.

Table 1: ID generalization scores for the 16-shot setup and $|\mathcal{D}_{\text{unlab}}| = 100$. The standard deviations of 10 runs are shown as subscripts. The highest scores and smallest standard deviations are highlighted in **bold**, while the second-best scores are underlined.

| | Method | GLUE | | | | | | | MMLU | |
|---|---|---|---|---|---|---|---|---|---|---|
| | | RTE | SST | QNLI | MNLI | COLA | MRPC | QQP | MATH | MISC |
| **Llama 3 (8B)** | 0-shot | 62.3 | 79.1 | 64.3 | 59.9 | 44.6 | 63.6 | 61.1 | 31.5 | 62.5 |
| | $n$-shot | $75.1_{6.5}$ | $93.5_{2.0}$ | $77.0_{5.5}$ | $68.0_{3.0}$ | $58.5_{4.0}$ | $74.0_{2.5}$ | $70.0_{3.0}$ | $43.5_{3.5}$ | $84.0_{4.0}$ |
| | PBFT | $73.2_{3.8}$ | $93.8_{1.5}$ | $77.8_{6.0}$ | $67.4_{3.5}$ | $56.5_{3.0}$ | $72.0_{2.0}$ | $68.0_{2.5}$ | $44.0_{3.8}$ | $83.5_{4.5}$ |
| | ICV | $72.9_{2.7}$ | $92.2_{1.8}$ | $74.5_{6.3}$ | $67.0_{4.2}$ | $57.3_{3.5}$ | $73.4_{2.3}$ | $69.1_{2.8}$ | $41.5_{4.3}$ | $67.0_{4.2}$ |
| | Batch-ICL | $77.8_{4.7}$ | $94.1_{2.2}$ | $78.0_{6.0}$ | $70.9_{3.5}$ | $59.8_{3.7}$ | $75.2_{2.2}$ | $\underline{72.5}_{2.7}$ | $36.2_{4.0}$ | $81.0_{2.5}$ |
| | WILDA-F | $83.4_{0.3}$ | $95.1_{\mathbf{0.6}}$ | $\underline{80.3}_{1.4}$ | $72.1_{2.5}$ | $\underline{63.7}_{\mathbf{1.5}}$ | $76.2_{1.8}$ | $71.9_{1.9}$ | $\underline{46.0}_{2.3}$ | $\underline{86.0}_{2.3}$ |
| | WILDA-S | $\underline{86.0}_{\mathbf{0.6}}$ | $\mathbf{96.1}_{1.2}$ | $\mathbf{81.4}_{2.2}$ | $\underline{73.1}_{\mathbf{2.0}}$ | $\mathbf{64.3}_{2.2}$ | $\mathbf{77.7}_{1.5}$ | $\mathbf{73.1}_{\mathbf{1.8}}$ | $\mathbf{49.5}_{2.0}$ | $\mathbf{88.0}_{2.2}$ |
| | WILDA-R | $\mathbf{86.5}_{3.0}$ | $\underline{95.5}_{0.8}$ | $79.0_{4.3}$ | $\mathbf{73.5}_{3.0}$ | $62.5_{2.8}$ | $\underline{76.5}_{1.9}$ | $72.0_{2.2}$ | $44.0_{2.7}$ | $85.5_{3.3}$ |
| **Phi 3 (mini 4k)** | 0-shot | 60.6 | 78.3 | 61.1 | 58.1 | 43.7 | 63.1 | 57.8 | 29.5 | 52.0 |
| | $n$-shot | $72.1_{5.2}$ | $90.6_{2.1}$ | $75.6_{3.2}$ | $65.3_{3.1}$ | $55.5_{4.1}$ | $71.1_{2.6}$ | $66.2_{3.7}$ | $37.5_{3.6}$ | $75.5_{4.1}$ |
| | PBFT | $70.6_{4.3}$ | $90.9_{1.9}$ | $73.6_{3.4}$ | $63.6_{3.6}$ | $53.6_{3.1}$ | $69.6_{2.3}$ | $64.6_{2.6}$ | $36.5_{4.1}$ | $73.5_{4.6}$ |
| | ICV | $71.5_{3.1}$ | $89.1_{2.1}$ | $74.3_{3.2}$ | $64.1_{4.1}$ | $54.1_{3.6}$ | $70.8_{2.4}$ | $65.4_{2.9}$ | $36.0_{4.6}$ | $74.0_{4.3}$ |
| | Batch-ICL | $75.3_{4.2}$ | $91.2_{2.6}$ | $76.6_{3.1}$ | $67.1_{3.6}$ | $56.1_{4.1}$ | $72.6_{2.6}$ | $67.3_{2.8}$ | $38.0_{3.9}$ | $76.0_{4.1}$ |
| | WILDA-F | $\underline{80.4}_{\mathbf{1.2}}$ | $92.1_{1.6}$ | $\underline{78.2}_{1.3}$ | $\underline{69.7}_{2.4}$ | $\underline{59.5}_{2.5}$ | $73.5_{2.1}$ | $\underline{68.6}_{2.2}$ | $\underline{40.5}_{3.2}$ | $\underline{77.5}_{3.6}$ |
| | WILDA-S | $\mathbf{82.4}_{1.1}$ | $\mathbf{93.2}_{1.6}$ | $79.2_{1.4}$ | $\mathbf{70.4}_{1.1}$ | $\mathbf{60.7}_{2.3}$ | $\mathbf{74.1}_{1.4}$ | $\mathbf{69.6}_{1.9}$ | $\mathbf{41.5}_{2.3}$ | $\mathbf{78.0}_{3.3}$ |
| | WILDA-R | $79.0_{1.9}$ | $\underline{92.6}_{2.0}$ | $\mathbf{79.6}_{2.9}$ | $68.6_{3.9}$ | $58.6_{2.9}$ | $\underline{73.6}_{2.0}$ | $68.1_{2.3}$ | $39.5_{3.6}$ | $77.0_{3.7}$ |

ICL (cf. Table 8 in the Appendix for other $n$-shot setups). Additionally, when a larger set $\mathcal{D}_{\text{unlab}}$ is used, there is a marginal improvement in scores, while stability improves even further (see Table 9 in the Appendix). Notably, the improvements in generalization with WILDA-S, compared to standard ICL (the teacher model in WILDA), provide strong evidence that the student model is exhibiting W2S generalization; we provide a more detailed analysis of this phenomenon in Section 4. While the WILDA-F and WILDA-R variants show similar generalization scores to WILDA-S, they typically exhibit higher variance. This makes WILDA-S the preferred choice due to its greater stability with respect to demonstration selection, as it consistently improves upon standard $n$-shot ICL across all datasets and models. This is supported by the statistically significant differences in standard deviations on all datasets for Llama 3 and on all but QNLI for Phi 3.[4]

Having looked at stability with respect to demonstration selection, we now turn to a more focused evaluation of stability with respect to demonstration ordering. Table 2 reports the standard deviations across 50 runs, where the same set of demonstrations is used, but their order is shuffled for each run. Designed to adapt to shuffled demonstrations, WILDA-S *shows the highest stability to demonstration ordering*, as evidenced by the smallest standard deviation. The stability improvements with WILDA-S over standard ICL are statistically significant across all datasets.[4]

We next assess the capacity of WILDA to perform OOD generalization by fine-tuning an adapter on one dataset and then applying the student model to a different dataset within the same task category, simulating a near-OOD scenario with pairs of closely related datasets. Table 3 shows the OOD generalization scores for such pairs of datasets in the GLUE benchmark. The results show that WILDA-S *not only outperforms other methods in OOD generalization but also maintains higher stability when adapting to new domains* (cf. Table 10 in the Appendix for results with other models).

Beyond generalization and stability, we also assess whether the adapters faithfully encode the information contained in the demonstrations. To evaluate this, we encode single demonstrations into the adapters and measure the student model's ability to reconstruct them using a simple recall task. As detailed in Appendix D.3, the adapters achieve consistently high semantic similarity across GLUE datasets, indicating that the demonstration content is reliably preserved. These findings corroborate the effectiveness of WILDA in capturing and *storing task-specific information within the adapter weights in a faithful and disentangled manner.*

---

[4] We test for significance using a two-tailed Levene's test ($p < 0.05$) and apply the Holm-Bonferroni method to correct for family-wise error rate.

Table 2: Standard deviations of generalization scores across 50 runs with varied orderings of 16 demonstrations. The smallest deviations are in **bold**, and the second-smallest are underlined.

| Model | Method | GLUE | | | | | | | MMLU | |
|---|---|---|---|---|---|---|---|---|---|---|
| | | **RTE** | **SST** | **QNLI** | **MNLI** | **COLA** | **MRPC** | **QQP** | **MATH** | **MISC** |
| Llama 3 (8B) | $n$-shot | 4.81 | 1.62 | 4.19 | 2.22 | 3.04 | 1.81 | 2.03 | 2.52 | 2.87 |
| | PBFT | 2.71 | 1.14 | 4.53 | 2.69 | 2.27 | 1.57 | 1.82 | 2.70 | 3.22 |
| | ICV | 2.09 | 1.23 | 4.08 | 2.81 | 1.95 | 1.61 | 2.03 | 1.96 | 3.18 |
| | Batch-ICL | 3.04 | 1.47 | 2.89 | 2.24 | 2.53 | 1.42 | 1.74 | 2.51 | 2.59 |
| | WILDA-F | 1.32 | 0.72 | 1.53 | 1.83 | 1.76 | 1.54 | 1.38 | 1.89 | 2.07 |
| | WILDA-S | **0.22** | **0.53** | **1.04** | **1.21** | **1.28** | **0.73** | **1.14** | **1.22** | **0.97** |
| | WILDA-R | 2.04 | 1.34 | 2.47 | 2.05 | 1.85 | 1.48 | 1.64 | 2.03 | 2.51 |

Table 3: OOD generalization scores with 16 shots averaged over 10 runs, with standard deviations shown as subscripts. For each dataset pair, demonstrations are taken from the **left** dataset, and the model is tested on the **right** dataset. Columns represent results on the **right** datasets. The highest scores and lowest standard deviations are in **bold**, and the second-highest scores are underlined. Values in parentheses indicate differences from ID performance for the corresponding target dataset.

| Model | Method | QNLI $\rightarrow$ RTE | RTE $\rightarrow$ QNLI | QQP $\rightarrow$ MRPC | MRPC $\rightarrow$ QQP |
|---|---|---|---|---|---|
| Llama 3 (8B) | $n$-shot | $66.3_{2.4}$ (8.8) | $69.6_{1.3}$ (7.4) | $66.5_{1.9}$ (7.5) | $62.2_{2.3}$ (7.8) |
| | PBFT | $66.1_{1.5}$ (7.1) | $69.1_{1.6}$ (8.7) | $67.2_{1.8}$ (4.8) | $62.4_{1.2}$ (5.6) |
| | ICV | $65.7_{1.2}$ (7.2) | $68.7_{2.3}$ (5.8) | $67.5_{1.6}$ (5.9) | $63.0_{2.1}$ (6.1) |
| | Batch-ICL | $65.3_{1.4}$ (12.5) | $66.3_{2.5}$ (11.7) | $64.9_{2.3}$ (10.3) | $62.1_{2.1}$ (10.4) |
| | WILDA-F | $\underline{67.5}_{1.1}$ (15.9) | $\underline{70.5}_{1.4}$ (9.8) | $68.5_{\mathbf{1.0}}$ (7.7) | $64.4_{1.5}$ (7.5) |
| | WILDA-S | $\mathbf{69.0}_{\mathbf{0.5}}$ (17.0) | $\mathbf{71.3}_{\mathbf{0.7}}$ (10.1) | $\mathbf{69.0}_{2.2}$ (8.7) | $\underline{66.4}_{\mathbf{1.1}}$ (6.7) |
| | WILDA-R | $67.1_{1.7}$ (19.4) | $70.0_{1.4}$ (9.0) | $68.0_{2.7}$ (8.5) | $\mathbf{68.3}_{2.0}$ (3.7) |

## 3.3 Adapter Arithmetic

To overcome the limitations of context window sizes and efficiently handle extensive demonstration sets in ICL, we employ *adapter arithmetic* within WILDA. This is achieved by fine-tuning separate adapters for each demonstration subset, with each adapter encoding the latent shift corresponding to its subset. Following the approach of Chitale et al. [6], these adapters are merged by summing their parameters, producing a single adapter that integrates knowledge from all subsets. Partitioning demonstrations into smaller subsets enables more effective use of the available context window, allowing models to incorporate more demonstrations without exceeding length limits or modifying the base LLM architecture. Additionally, distributing the prompt across multiple adapters improves GPU utilization by fitting it on a single GPU and reducing memory overhead during inference.

Table 4 shows the ID generalization scores of ICV, Batch-ICL, and WILDA in fusing knowledge from multiple demonstration subsets, specifically using 2, 4, and 8 subsets of 16 demonstrations each. WILDA-S consistently outperforms baseline methods, highlighting its effectiveness in knowledge fusion across subsets [36]. Moreover, this form of adapter arithmetic aligns with recent advances in task arithmetic, where merging task-specific parameters promotes generalization across multiple tasks [19, 31]. In our case, *this approach effectively improves generalization and stability when fusing demonstration subsets within the same task.*

## 3.4 Few-Shot WILDA

The core idea behind WILDA is to remove the need for explicit demonstrations by encoding their effect directly into the adapter. However, we also want to verify that standard ICL remains functional when the adapter is active. In particular, we examine whether the latent shift captured by the adapter composes additively with the contextual shift induced by new demonstrations provided during inference.

To this end, we evaluate WILDA-S using Llama 3 (8B) in a mixed few-shot configuration. Specifically, we consider a setup in which the adapter has been fine-tuned to encode 16 demonstrations and is further provided with an additional 16 in-context demonstrations during inference. This configuration,

Table 4: ID generalization scores of knowledge fusion for Llama 3 (8B). The scores are averaged over 10 runs with standard deviations shown as subscripts. The table compares the effectiveness of knowledge fusion from 2, 4, and 8 subsets of 16 demonstrations. The highest scores are in **bold**.

| Demonstrations | Method | GLUE | | | | | | | MMLU | |
|---|---|---|---|---|---|---|---|---|---|---|
| | | RTE | SST | QNLI | MNLI | COLA | MRPC | QQP | MATH | MISC |
| $2 \times 16$ | ICV | $75.2_{4.3}$ | $93.6_{1.9}$ | $77.6_{5.9}$ | $69.2_{3.7}$ | $58.3_{3.5}$ | $74.2_{2.4}$ | $70.6_{2.7}$ | $45.5_{3.7}$ | $72.5_{2.9}$ |
| | Batch-ICL | $80.2_{3.6}$ | $95.3_{1.8}$ | $80.2_{5.8}$ | $72.3_{3.0}$ | $61.2_{3.1}$ | $76.3_{2.0}$ | $72.6_{2.4}$ | $43.5_{2.9}$ | $83.0_{3.6}$ |
| | WILDA-S | $\mathbf{87.1}_{1.6}$ | $\mathbf{96.4}_{1.3}$ | $\mathbf{81.5}_{5.0}$ | $\mathbf{75.5}_{2.5}$ | $\mathbf{68.4}_{1.8}$ | $\mathbf{78.5}_{1.4}$ | $\mathbf{74.1}_{1.6}$ | $\mathbf{51.5}_{1.6}$ | $\mathbf{89.5}_{2.0}$ |
| $4 \times 16$ | ICV | $78.3_{3.6}$ | $94.6_{1.8}$ | $79.3_{5.5}$ | $71.2_{3.1}$ | $60.3_{3.3}$ | $75.6_{2.2}$ | $72.3_{2.4}$ | $47.5_{3.5}$ | $76.5_{3.8}$ |
| | Batch-ICL | $84.4_{3.3}$ | $96.4_{1.5}$ | $82.4_{5.2}$ | $74.3_{2.5}$ | $64.2_{2.8}$ | $78.3_{1.6}$ | $74.3_{2.1}$ | $45.5_{2.6}$ | $84.5_{3.3}$ |
| | WILDA-S | $\mathbf{88.4}_{2.3}$ | $\mathbf{97.5}_{0.7}$ | $\mathbf{83.6}_{4.4}$ | $\mathbf{77.3}_{2.2}$ | $\mathbf{71.4}_{1.5}$ | $\mathbf{79.6}_{0.7}$ | $\mathbf{75.2}_{1.3}$ | $\mathbf{53.5}_{1.4}$ | $\mathbf{91.0}_{1.7}$ |
| $8 \times 16$ | ICV | $81.3_{2.8}$ | $95.6_{1.5}$ | $81.8_{5.0}$ | $73.3_{2.7}$ | $61.3_{2.4}$ | $77.3_{1.7}$ | $73.8_{2.0}$ | $47.5_{2.9}$ | $78.0_{3.5}$ |
| | Batch-ICL | $85.6_{2.5}$ | $96.7_{1.1}$ | $83.8_{4.5}$ | $75.8_{2.1}$ | $65.3_{2.1}$ | $79.8_{1.3}$ | $75.8_{1.8}$ | $45.5_{2.0}$ | $84.0_{2.5}$ |
| | WILDA-S | $\mathbf{92.8}_{0.8}$ | $\mathbf{98.1}_{0.2}$ | $\mathbf{87.9}_{2.5}$ | $\mathbf{81.3}_{0.9}$ | $\mathbf{74.1}_{0.6}$ | $\mathbf{82.8}_{0.4}$ | $\mathbf{78.9}_{0.5}$ | $\mathbf{57.0}_{0.5}$ | $\mathbf{93.0}_{0.7}$ |

Table 5: Performance comparison of WILDA-S configurations and standard 32-shot ICL using Llama 3 (8B). Results are averaged over 10 runs. The notation $n/d$ indicates the number of demonstrations in context ($n$) and encoded in the adapter ($d$).

| Method | GLUE | | | | | | | MMLU | |
|---|---|---|---|---|---|---|---|---|---|
| | RTE | SST | QNLI | MNLI | COLA | MRPC | QQP | MATH | MISC |
| 32-shot ICL | 75.3 | 93.2 | 77.7 | 69.1 | 58.3 | 76.4 | 74.2 | 43.0 | 84.5 |
| WILDA-S (0/32) | 87.9 | 97.9 | 83.1 | 74.0 | 64.6 | 79.4 | 74.8 | 56.5 | 89.0 |
| WILDA-S (0/16) | 86.0 | 96.1 | 81.4 | 73.1 | 64.3 | 77.7 | 73.1 | 49.5 | 88.0 |
| WILDA-S (16/16) | 87.3 | 96.4 | 82.2 | 74.6 | 65.4 | 78.2 | 74.5 | 51.0 | 89.0 |

denoted as $16/16$, thus combines parameter-based and context-based adaptation. We compare it against several baselines: standard 32-shot ICL, WILDA-S with 16 encoded demonstrations and no in-context examples ($0/16$), and WILDA-S with 32 encoded demonstrations and no in-context examples ($0/32$).

The results, summarized in Table 5, show that the hybrid $16/16$ configuration outperforms standard 32-shot ICL across all evaluated datasets, indicating that the adapter and in-context demonstrations reinforce one another rather than interfering. While $16/16$ setup performs slightly below the fully encoded $0/32$ variant, likely because the latter benefits from a dedicated fine-tuning phase, the $16/16$ setup demonstrates that WILDA can successfully integrate additional ICL prompts on top of a fine-tuned adapter. This finding confirms that standard ICL remains fully compatible with adapter-based tuning, enabling a composition of contextual and parametric adaptation mechanisms within a single framework.

## 4 Analysis of Weak-to-Strong Generalization

Building on the observation that WILDA consistently outperforms its teacher, standard ICL, we hypothesize that W2S generalization may be driving these improvements, where the model's ability to generalize strengthens progressively from weaker signals. To explore this further, we conduct an empirical analysis of WILDA-S with Llama 3 on aggregated examples from all GLUE datasets, treating them as a single, unified dataset. We focus on the WILDA-S variant due to its consistently strong performance and stability across prior experiments.

A crucial prerequisite for successful W2S generalization is the student's ability to maintain stable outputs under small perturbations of the input, i.e., robustness to input variations. A low Lipschitz constant serves as a key indicator of this stability, as it bounds the maximum change in the model output for any change in its input [21]. However, calculating the exact Lipschitz constant for LLMs is intractable. To approximate it, we leverage the relationship between the Lipschitz constant and the input–output Jacobian matrix of a neural network. Specifically, we compute the Frobenius norm of the Jacobian matrix as a tractable proxy, given its relationship to the spectral norm, which is a known lower bound for the Lipschitz constant [9] (see Appendix B). Figure 2a shows a histogram of the

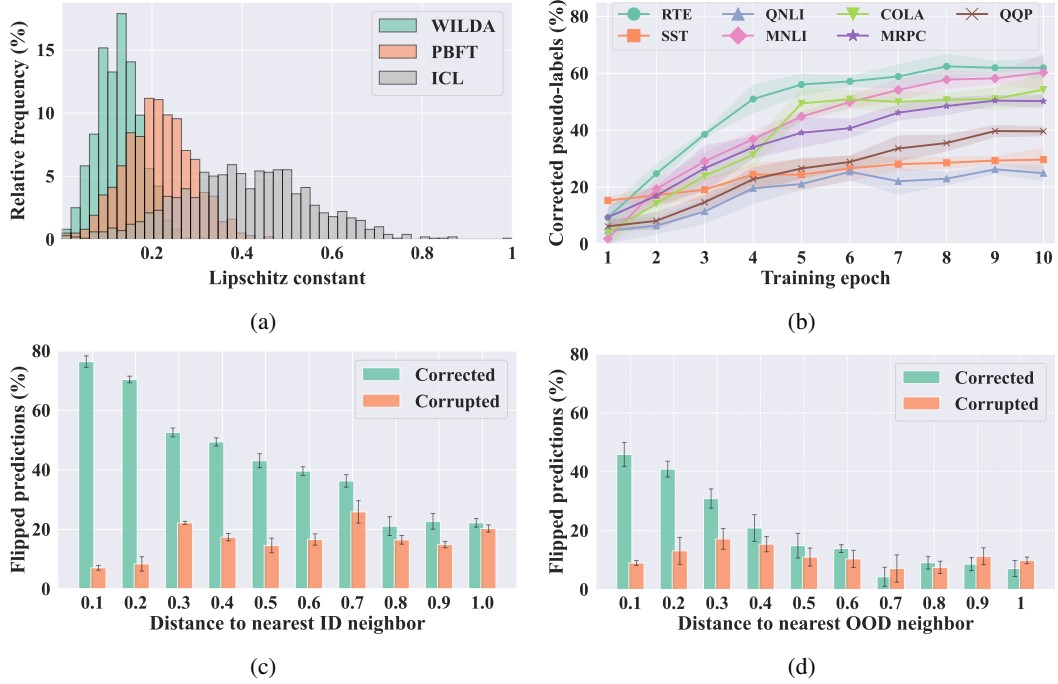

(a)

(b)

(c)

(d)

Figure 2: Empirical analysis of WILDA-S on the aggregated GLUE datasets for Llama 3: (a) **Histogram of approximated Lipschitz constants** across datasets, computed as the Frobenius norm of the input–output Jacobian matrix; (b) **Rate of pseudo-label correction** over training epochs (shaded areas indicate the standard deviation over 10 runs); **Corrected and corrupted prediction rates** for (c) **ID examples** and (d) **OOD examples**, based on the normalized Euclidean distance to the nearest correctly pseudo-labeled neighbor. Error bars indicate standard deviation over 10 runs.

approximated Lipschitz constants (normalized to $[0, 1]$) for WILDA, PBFT, and ICL. WILDA exhibits a notably lower Lipschitz constant than PBFT and ICL, which reflects its stronger local consistency.

The student's ability to revise the teacher's predicted labels, known as *pseudo-label correction*, builds directly on local consistency and serves as a key mechanism underlying W2S generalization [23]. When the model exhibits stable behavior under small input perturbations, as captured by a low Lipschitz constant, it is more likely to propagate corrections reliably across neighboring inputs in representation space. This local consistency forms the foundation for accurate correction of noisy pseudo-labels in high-confidence regions. As corrected labels accumulate, they create a foundation for generalization to nearby, low-confidence examples – gradually expanding the model's coverage and facilitating a transition from local consistency to broader generalization. Figure 2b shows how the rate of corrected pseudo-labels evolves during training on GLUE datasets. As training progresses, the percentage of corrected pseudo-labels steadily increases, demonstrating WILDA's capacity to exhibit W2S generalization. Notably, the rate of pseudo-label correction plateaus faster for simpler datasets like SST and QNLI, which have lower linguistic variability.

The mechanism of pseudo-label correction ties into the phenomenon of *coverage expansion*, where the model generalizes beyond the regions covered by pseudo-labels [23]. We hypothesize that WILDA's generalization ability is supported by coverage expansion, where local corrections gradually influence nearby examples in representation space. The model's coverage expands incrementally through a ripple effect, with high-confidence predictions influencing nearby examples while remaining grounded in regions supported by learned corrections. To understand this dynamic, we analyze which unseen evaluation points are corrected by clustering them based on their proximity to the nearest correctly pseudo-labeled neighbor in $\mathcal{D}_{\text{unlab}}$. This is quantified by computing the Euclidean distance between the model's representations at the final hidden states, with evaluation points categorized into ten bins based on their normalized distance from the correct neighbor. Figure 2c illustrates the rate of prediction flips within these bins, where a flip refers to either correcting an incorrect prediction or corrupting a correct one. The rate of corrected predictions shows a strong negative

association with the distance to the nearest correctly labeled neighbor, as measured by the point biserial correlation coefficient of $-0.968$. In contrast, corrupted predictions are more frequent in regions lacking nearby correct pseudo-labels. Moreover, coverage expansion also shows its effects on OOD data. Figure 2d shows the rate of flipped predictions for OOD data. Although the impact is reduced, a similar correction pattern persists, with a point biserial correlation of $-0.916$.

Overall, the results indicate that WILDA does not merely replicate the teacher's outputs but progressively reorganizes its representation space to reinforce locally consistent decision regions. This smoothing effect allows reliable predictions to extend beyond areas directly supported by demonstrations, consolidating weak contextual supervision into more stable and transferable task knowledge.

## 5   Related Work

Recent perspectives on ICL have shifted from task learning to task identification. Wies et al. [40] argue that ICL works by recognizing latent tasks embedded during pre-training. Hoogland et al. [15] build on this, suggesting that ICL unfolds in developmental stages, shedding light on how models adapt to novel contexts. Li et al. [26] further empirically show that ICL predictions become more resilient to input perturbations with longer prompts and that training on noisy data enhances stability. Despite these theoretical breakthroughs, ICL remains vulnerable to the selection and ordering of demonstrations [25, 30]. Moreover, Kossen et al. [22] highlight ICL's biases rooted in pre-training data, revealing that models do not always uniformly leverage in-context information.

Research into the inner workings of ICL has revealed how transformers process demonstrations to form task representations. Hendel et al. [13] and Liu et al. [28] show that transformers can compress demonstration examples into a task vector, which efficiently directs the model to generate context-appropriate outputs for queries. These task vectors are created during a forward pass, capturing the latent shift induced by the demonstrations. Building on this, Dai et al. [8] explore using linear attention to compute virtual gradients, simulating the effect of gradient-based learning within the model. Similarly, Todd et al. [34] use causal mediation analysis to highlight the role of specific attention heads in forming robust task representations in ICL, termed function vectors.

Wei et al. [39] provide a theoretical foundation for W2S generalization, showing that, under the assumption of coverage expansion, models optimized for population-level consistency can achieve high accuracy. Lang et al. [23] further advance this view by formalizing the role of pseudo-label correction, which emerges when a model enforces local consistency during training. Building on these principles, several recent works demonstrate how large language models (LLMs) can leverage their own high-confidence outputs to improve performance. For instance, Huang et al. [18] show that rationale-augmented predictions can guide fine-tuning and enhance reasoning abilities without labeled supervision. Similarly, Qu et al. [32] propose recursive introspection for iterative improvement, and Wang et al. [38] introduce self-taught evaluators that enable LLMs to refine their outputs over time.

## 6   Conclusion

To tackle the challenges of stability and long-context handling that arise when processing multiple demonstrations in ICL within LLMs, we introduced WILDA, a method that disentangles the latent shifts induced by demonstrations from those of the query, leveraging a teacher–student framework. WILDA encodes these latent shifts into an adapter module, enabling the student model to handle queries without requiring demonstrations in the input. Moreover, WILDA allows efficient handling of large demonstration sets by chunking them into manageable subsets, each processed through separate adapter modules. This not only reduces the instability caused by demonstration selection and ordering but also alleviates the context window limitations inherent in transformer-based models. Additionally, we demonstrated that WILDA exhibits weak-to-strong generalization by refining pseudo-labels through progressive corrections, expanding from local consistency to a more comprehensive coverage across the representation space. Our empirical evaluation confirms these advantages, showing that WILDA consistently outperforms traditional ICL methods, significantly improving generalization and stability across diverse natural language understanding datasets. Ultimately, by treating ICL as weak supervision, WILDA enables parametric generalization that can flexibly incorporate contextual adaptation, turning transient in-context behaviors into persistent, reusable parameters and providing a scalable and modular foundation for stable task adaptation in LLMs.

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

# A  Dual Form of ICL

We offer a detailed derivation of (3), originally introduced by [8], expanding on the key intermediate steps for clarity, which were not explicitly covered in the original work. The goal is to decompose the attention head output into separate components corresponding to the demonstrations and the query, thereby disentangling the latent shifts induced by ICL.

## A.1  Linearized Attention Formulation

We begin with the approximation of the attention head's output using linear attention:

$$\mathbf{f}_{\text{AH}}(\mathbf{x}_q^{(t)}) \approx \mathbf{W}_V[\mathbf{X}_d; \mathbf{X}_q] \left(\mathbf{W}_K[\mathbf{X}_d; \mathbf{X}_q]\right)^{\top} \mathbf{q}^{(t)}, \tag{7}$$

where:

- $\mathbf{W}_V \in \mathbb{R}^{d_h \times d_{\text{model}}}$ is the value weight matrix;
- $\mathbf{W}_K \in \mathbb{R}^{d_h \times d_{\text{model}}}$ is the key weight matrix;
- $\mathbf{X}_d \in \mathbb{R}^{d_{\text{model}} \times N_d}$ is the matrix of demonstration token representations;
- $\mathbf{X}_q \in \mathbb{R}^{d_{\text{model}} \times N_q}$ is the matrix of previous query token representations up to time $t-1$;
- $\mathbf{q}^{(t)} = \mathbf{W}_Q \mathbf{x}_q^{(t)} \in \mathbb{R}^{d_h}$ is the query vector at time $t$, with $\mathbf{W}Q \in \mathbb{R}^{d_h \times d_{\text{model}}}$ being the query weight matrix;
- $[\mathbf{X}_d; \mathbf{X}_q]$ is the concatenation of $\mathbf{X}_d$ and $\mathbf{X}_q$ along the sequence dimension.

## A.2  Expanding the Concatenated Matrices

We can expand the concatenated matrices as follows:

$$\mathbf{W}_V[\mathbf{X}_d; \mathbf{X}_q] = [\mathbf{W}_V\mathbf{X}_d; \mathbf{W}_V\mathbf{X}_q] = [\mathbf{V}_d; \mathbf{V}_q], \tag{8}$$
$$\mathbf{W}_K[\mathbf{X}_d; \mathbf{X}_q] = [\mathbf{W}_K\mathbf{X}_d; \mathbf{W}_K\mathbf{X}_q] = [\mathbf{K}_d; \mathbf{K}_q], \tag{9}$$

where:

- $\mathbf{V}_d = \mathbf{W}_V\mathbf{X}_d$ is the value matrix for the demonstrations;
- $\mathbf{V}_q = \mathbf{W}_V\mathbf{X}_q$ is the value matrix for the previous queries;
- $\mathbf{K}_d = \mathbf{W}_K\mathbf{X}_d$ is the key matrix for the demonstrations;
- $\mathbf{K}_q = \mathbf{W}_K\mathbf{X}_q$ is the key matrix for the previous queries.

The transpose of the concatenated key matrix is:

$$\left(\mathbf{W}_K[\mathbf{X}_d; \mathbf{X}_q]\right)^{\top} = \left[\mathbf{K}_d^{\top}; \mathbf{K}_q^{\top}\right]. \tag{10}$$

## A.3  Block Matrix Multiplication

Substituting the expanded forms into Equation (7) using rules for block matrix multiplication, we have:
$$\mathbf{f}_{\text{AH}}(\mathbf{x}_q^{(t)}) \approx [\mathbf{V}_d; \mathbf{V}_q] \left[\mathbf{K}_d^{\top}; \mathbf{K}_q^{\top}\right] \mathbf{q}^{(t)} = \left(\mathbf{V}_d\mathbf{K}_d^{\top} + \mathbf{V}_q\mathbf{K}_q^{\top}\right) \mathbf{q}^{(t)}. \tag{11}$$
This separates the contributions from the demonstrations and the query sequences.

## A.4  Component Definitions

We define:

$$\mathbf{W}_{\text{ZS}} = \mathbf{V}_q\mathbf{K}_q^{\top} = \mathbf{W}_V\mathbf{X}_q \left(\mathbf{W}_K\mathbf{X}q\right)^{\top}, \tag{12}$$
$$\Delta\mathbf{W}_{\text{ICL}} = \mathbf{V}_d\mathbf{K}_d^{\top} = \mathbf{W}_V\mathbf{X}_d \left(\mathbf{W}_K\mathbf{X}_d\right)^{\top}. \tag{13}$$

Here:

- $\mathbf{W}_{\mathrm{ZS}}$ represents the zero-shot component, capturing the model's behavior based on the query sequence alone;
- $\Delta\mathbf{W}_{\mathrm{ICL}}$ represents the latent shift induced by the demonstrations, capturing the effect of in-context learning.

## A.5 Final Decomposition

Substituting (12) and (13) back into the expression, we obtain:

$$\mathbf{f}_{\mathrm{AH}}(\mathbf{x}_q^{(t)}) \approx \left(\mathbf{W}_{\mathrm{ZS}} + \Delta\mathbf{W}_{\mathrm{ICL}}\right)\mathbf{q}^{(t)} = \mathbf{W}_{\mathrm{ZS}}\mathbf{q}^{(t)} + \Delta\mathbf{W}_{\mathrm{ICL}}\mathbf{q}^{(t)}. \tag{14}$$

The decomposition shows that the attention head output can be viewed as the sum of:

1. The **zero-shot component** ($\mathbf{W}_{\mathrm{ZS}}\mathbf{q}^{(t)}$): the model's output when only the query sequence is considered, without any influence from the demonstrations;
2. The **latent shift due to ICL** ($\Delta\mathbf{W}_{\mathrm{ICL}}\mathbf{q}^{(t)}$): the additional contribution from the demonstrations, representing the knowledge introduced via in-context learning.

This separation aligns with the theoretical motivation to disentangle the latent shifts induced by the demonstrations from those induced by the query, allowing for more efficient and stable processing of queries independently of demonstrations.

# B  Lipschitz Continuity in Neural Networks

Lipschitz continuity is a fundamental concept in the analysis of neural networks as it provides a bound on how much the output of a function can change with respect to its input. Formally, a function $f : \mathbb{R}^n \to \mathbb{R}^m$ is said to be Lipschitz continuous with constant $L \geq 0$ if for any two inputs $\mathbf{x}, \mathbf{x}' \in \mathbb{R}^n$ the following inequality holds:

$$\|f(\mathbf{x}) - f(\mathbf{x}')\| \leq L\|\mathbf{x} - \mathbf{x}'\|.$$

This property ensures that the function $f$ behaves smoothly, meaning small changes in the input lead to small changes in the output, which is crucial for robustness in neural networks, particularly for predictive models [21].

## B.1  Relationship Between the Lipschitz Constant and the Jacobian Matrix

In neural networks, the Lipschitz constant can be bounded by the spectral norm of the Jacobian matrix, which quantifies the sensitivity of a function's output to changes in the input. The Jacobian matrix $\mathbf{J}_f(\mathbf{x}) \in \mathbb{R}^{m \times n}$ of a function $f$ is defined as the matrix of all partial derivatives:

$$[\mathbf{J}_f(\mathbf{x})]_{i,j} = \frac{\partial f_i(\mathbf{x})}{\partial x_j}.$$

The spectral norm of the Jacobian matrix, denoted $\|\mathbf{J}_f(\mathbf{x})\|_2$, provides a pointwise lower bound on the global Lipschitz constant $L$:

$$\|\mathbf{J}_f(\mathbf{x})\|_2 \leq L, \forall \mathbf{x} \in \mathbb{R}^n.$$

The spectral norm represents the greatest possible rate of change in the function's output for any input variation. However, calculating the exact spectral norm can be computationally expensive, especially for deep neural networks, so the Frobenius norm is often used as an efficient alternative.

## B.2  Frobenius Norm as a Surrogate for the Lipschitz Constant

The Frobenius norm of the Jacobian matrix is often used as a surrogate for estimating the Lipschitz constant to avoid the computational complexity of calculating the spectral norm. The Frobenius norm, denoted $\|\mathbf{A}\|_F$, is easier to compute and relates to the spectral norm through the following inequality:

$$\|\mathbf{A}\|_2 \leq \|\mathbf{A}\|_F \leq \sqrt{r}\|\mathbf{A}\|_2,$$

where $r$ is the rank of the matrix $\mathbf{A}$. While the Frobenius norm generally overestimates the spectral norm, the bounded gap between them implies that reducing the Frobenius norm below its initial value often corresponds to a decrease in the spectral norm as well. This relationship supports its use as a practical proxy for Lipschitz behavior: reductions in the Frobenius norm are generally associated with a lower Lipschitz constant, supported by evidence showing that the Lipschitz constant of neural networks tends to closely track the lower bound defined by the spectral norm [24, 21].

### B.3 Empirical Evaluation of Lipschitz Continuity

In our experiments, we approximate the Lipschitz constant by computing the Frobenius norm of the input–output Jacobian matrix, where the embeddings are the inputs and the penultimate layer produces the outputs. As shown in Figure 2a, WILDA demonstrates a significantly lower approximated Lipschitz constant compared to PBFT and ICL. This lower value suggests that WILDA is more robust to input perturbations, which is a critical property for correcting pseudo-labels.

## C  Limitations

**Computational cost.**  WILDA introduces additional computational overhead due to the fine-tuning of adapters. While this fine-tuning is more lightweight compared to full model fine-tuning, it remains more expensive than standard ICL, which avoids weight updates entirely. However, WILDA offsets some of this cost by removing demonstrations from the input during inference. For instance, with Llama 3 (8B) processing 16 demonstrations from GLUE datasets, inference takes approximately 120 times longer than a 0-shot setup (processing only the query). This increased cost scales quadratically with the number of tokens, highlighting the self-attention mechanism as the primary bottleneck when handling 16 demonstrations. Based on our measurements, fine-tuning with 100 unlabeled instances and 16 demonstrations using a single adapter corresponds to the computational cost of approximately 2100 inferences in a 16-shot setup. This implies that after about 2100 inferences, the time spent on fine-tuning is effectively balanced by the reduction in per-inference computational cost.

**Applicability.**  WILDA may be less suitable for scenarios with extremely limited resources, as it relies on access to a supply of unlabeled data. In our experiments with $\{4, 8, 16, 32\}$ demonstrations, we typically used 100 unlabeled instances, which proved sufficient to achieve strong performance. While unlabeled data is generally easier to acquire than labeled data, there may be scenarios where obtaining even a modest amount of unlabeled data is challenging, potentially limiting the applicability of WILDA.

**Large demonstration sets.**  Although WILDA efficiently encodes demonstrations into adapters to overcome context length limitations, the method has not been extensively tested with very large demonstration sets. From our findings, as the total number of demonstrations increases, using multiple adapters with manageable demonstration sizes tends to be more effective. For instance, we successfully employed 8 adapters with 16 demonstrations each (totaling 128 demonstrations). While this approach theoretically allows for an indefinite increase in the number of demonstrations, its effectiveness with significantly larger sets remains unexplored. Moreover, using additional adapters increases computational costs, introducing a tradeoff between scalability and efficiency.

## D  Additional Results

### D.1  Supplementary ID and OOD Experiments

We provide additional experimental results that complement the analyses in the main paper. Table 6 presents extended in-domain generalization results under the 16-shot setup with $|\mathcal{D}_{\text{unlab}}| = 100$, confirming that WILDA consistently outperforms standard ICL and related baselines across GLUE and MMLU benchmarks. We further evaluate on the ARC-Challenge benchmark (Table 7), where WILDA-S attains the strongest accuracy, demonstrating that the observed gains extend to more demanding reasoning tasks. Table 8 examines the effect of the number of demonstrations ($n \in \{4, 8, 32\}$), showing that WILDA-S scales predictably with $n$ and remains markedly more data-efficient than standard ICL. We also assess the influence of the unlabeled query set size (Table 9), observing steady yet saturating improvements as $|\mathcal{D}_{\text{unlab}}|$ increases from 200 to 500. Finally, Table 10 summarizes

Table 6: ID generalization scores for the 16-shot scenario and $|\mathcal{D}_{\text{unlab}}| = 100$ for Llama 2 (7B). The standard deviations of 10 runs are shown as subscripts.

| Model | Method | GLUE | | | | | | | MMLU | |
|---|---|---|---|---|---|---|---|---|---|---|
| | | RTE | SST | QNLI | MNLI | COLA | MRPC | QQP | MATH | MISC |
| Llama 2 (7B) | 0-shot | 57.8 | 75.4 | 59.3 | 55.7 | 40.7 | 59.4 | 58.7 | 29.0 | 59.0 |
| | $n$-shot | $69.2_{4.3}$ | $89.8_{2.1}$ | $74.2_{5.9}$ | $63.3_{2.8}$ | $54.3_{3.5}$ | $66.9_{2.4}$ | $64.7_{1.5}$ | $37.5_{4.8}$ | $80.0_{5.3}$ |
| | PBFT | $69.0_{2.7}$ | $89.7_{0.4}$ | $73.3_{5.0}$ | $64.4_{4.7}$ | $51.2_{2.9}$ | $67.9_{2.0}$ | $64.6_{1.6}$ | $40.0_{3.2}$ | $79.5_{2.1}$ |
| | ICV | $68.0_{4.6}$ | $87.8_{2.6}$ | $71.2_{6.7}$ | $60.9_{4.0}$ | $53.1_{2.4}$ | $68.8_{1.7}$ | $65.0_{1.9}$ | $39.5_{2.7}$ | $62.5_{0.6}$ |
| | Batch-ICL | $75.2_{0.8}$ | $91.2_{1.9}$ | $74.0_{0.8}$ | $66.5_{3.3}$ | $55.9_{2.1}$ | $70.3_{0.8}$ | $69.1_{1.8}$ | $34.5_{2.3}$ | $77.0_{4.1}$ |
| | WILDA-F | $77.2_{0.7}$ | $90.2_{0.7}$ | $76.8_{4.2}$ | $66.5_{2.4}$ | $60.1_{1.2}$ | $71.6_{0.2}$ | $68.8_{0.8}$ | $43.0_{1.6}$ | $82.5_{2.5}$ |
| | WILDA-S | $81.9_{2.5}$ | $92.1_{0.3}$ | $77.3_{0.9}$ | $70.4_{1.8}$ | $62.8_{3.4}$ | $72.3_{2.6}$ | $68.2_{0.5}$ | $46.5_{1.5}$ | $82.5_{1.7}$ |
| | WILDA-R | $81.1_{1.9}$ | $93.6_{2.0}$ | $74.7_{3.6}$ | $69.6_{2.9}$ | $57.9_{2.9}$ | $73.1_{2.0}$ | $66.8_{2.3}$ | $41.5_{2.6}$ | $82.0_{3.7}$ |

Table 7: Accuracy on the ARC-Challenge benchmark using Llama 3 (8B) with $|\mathcal{D}_{\text{unlab}}| = 100$ and $n = 16$ shots. The highest score is shown in **bold**, and the second-best score is underlined.

| Method | 0-shot | $n$-shot | PBFT | ICV | Batch-ICL | WILDA-F | WILDA-S | WILDA-R |
|---|---|---|---|---|---|---|---|---|
| **Accuracy** | 38.3 | 54.5 | 50.8 | 49.9 | 48.6 | 55.7 | **59.2** | 56.5 |

out-of-domain generalization results, indicating that WILDA preserves strong transfer and stability across heterogeneous evaluation settings.

## D.2 Task Specificity and Generalization

LoRA adapters are implemented as additive residuals and do not alter the underlying model parameters. During training, we alternate between enabling the adapter (student) and disabling it (teacher), while at inference the adapter can be toggled on or off without modifying the base model. To assess whether task-specific adaptation affects general performance, we conducted additional experiments evaluating the model on unrelated, non-overlapping dataset pairs from GLUE. In each case, both 16-shot ICL and WILDA-S were exposed to demonstrations from a different domain. Results, summarized in Table 11, show that performance differences remain within one standard deviation across 10 random seeds. This indicates that WILDA-S does not degrade the model's general zero-shot capabilities, even when an adapter trained on a specific task remains active during evaluation on a disjoint domain.

## D.3 Faithful Encoding and Retrieval of Demonstrations

To evaluate whether demonstrations are faithfully encoded and disentangled, we conducted an experiment by encoding a single demonstration into the adapter and assessing the student model's ability to capture this information. Specifically, we utilized 1000 examples per dataset across the GLUE benchmark using Llama 3 (8B).

For each dataset, the student model was prompted with a simple instruction: "Repeat the demonstration word for word." During the fine-tuning phase, the teacher model processed input examples

Table 8: ID generalization scores for $n$-shot scenarios ($n = 4, 8, 32$, with $|\mathcal{D}_{\text{unlab}}| = 100$) for Llama 3 (8B). The standard deviations of 10 runs are shown as subscripts.

| Model | $n$ | Method | GLUE | | | | | | | MMLU | |
|---|---|---|---|---|---|---|---|---|---|---|---|
| | | | RTE | SST | QNLI | MNLI | COLA | MRPC | QQP | MATH | MISC |
| Llama 3 (8B) | 4 | $n$-shot | $71.3_{5.4}$ | $84.5_{4.4}$ | $70.1_{2.9}$ | $62.4_{2.7}$ | $54.6_{3.5}$ | $69.2_{4.1}$ | $62.0_{2.3}$ | $37.0_{3.9}$ | $76.5_{2.5}$ |
| | | WILDA-S | $80.3_{1.5}$ | $90.9_{0.9}$ | $76.3_{1.4}$ | $70.1_{1.8}$ | $61.4_{2.0}$ | $72.9_{1.5}$ | $70.3_{1.2}$ | $43.0_{1.3}$ | $77.5_{1.8}$ |
| | 8 | $n$-shot | $72.7_{2.1}$ | $89.4_{2.6}$ | $73.5_{2.5}$ | $64.7_{3.1}$ | $55.8_{2.8}$ | $71.2_{2.4}$ | $64.3_{2.9}$ | $37.0_{1.3}$ | $77.5_{2.1}$ |
| | | WILDA-S | $82.1_{1.1}$ | $93.2_{1.0}$ | $78.3_{1.3}$ | $72.2_{1.6}$ | $63.7_{1.8}$ | $73.9_{1.3}$ | $72.1_{0.4}$ | $47.5_{0.5}$ | $84.0_{1.4}$ |
| | 32 | $n$-shot | $75.3_{3.2}$ | $93.2_{1.9}$ | $77.7_{2.9}$ | $69.1_{1.9}$ | $58.3_{1.5}$ | $76.4_{2.2}$ | $74.2_{1.9}$ | $43.0_{1.5}$ | $84.5_{2.1}$ |
| | | WILDA-S | $87.9_{0.6}$ | $97.9_{0.4}$ | $83.1_{0.9}$ | $74.0_{1.1}$ | $64.6_{1.2}$ | $79.4_{0.6}$ | $74.8_{1.5}$ | $56.5_{0.2}$ | $89.0_{0.4}$ |

Table 9: ID generalization scores of WILDA-S for $n = 16$ shots and $|\mathcal{D}_{\text{unlab}}| = 200, 500$ for Llama 3 (8B). Results are shown for GLUE datasets with $n$-shot and WILDA-S methods. The standard deviations of 10 runs are shown as subscripts.

| Model | $|\mathcal{D}_{\text{unlab}}|$ | GLUE | | | | | | |
|---|---|---|---|---|---|---|---|---|
| | | RTE | SST | QNLI | MNLI | COLA | MRPC | QQP |
| Llama 3 (8B) | 200 | $86.2_{0.4}$ | $97.2_{0.4}$ | $81.6_{1.0}$ | $73.9_{1.3}$ | $64.7_{1.1}$ | $78.9_{0.7}$ | $74.0_{0.5}$ |
| | 500 | $86.9_{0.3}$ | $97.1_{0.5}$ | $81.9_{0.7}$ | $74.8_{1.0}$ | $64.6_{0.8}$ | $81.4_{0.8}$ | $75.2_{0.3}$ |

Table 10: OOD generalization scores for Phi 3 and Llama 2 in a 16-shot scenario with $\mathcal{D}_{\text{unlab}} = 100$ over 10 runs with standard deviations shown as subscripts. In each dataset pair, demonstrations are taken from the left dataset, and the model is tested on the right dataset. The columns correspond to the results on the right datasets.

| Model | Method | QNLI $\rightarrow$ RTE | RTE $\rightarrow$ QNLI | QQP $\rightarrow$ MRPC | MRPC $\rightarrow$ QQP |
|---|---|---|---|---|---|
| Phi 3 (mini 4k) | $n$-shot | $64.3_{2.5}$ | $67.2_{1.5}$ | $63.7_{2.3}$ | $59.4_{2.2}$ |
| | PBFT | $64.1_{1.8}$ | $66.9_{1.6}$ | $64.7_{2.0}$ | $60.1_{1.4}$ |
| | WILDA-S | $67.4_{0.6}$ | $69.2_{0.9}$ | $66.3_{2.4}$ | $64.4_{1.3}$ |
| Llama 2 (7B) | $n$-shot | $62.9_{2.3}$ | $66.3_{1.2}$ | $64.5_{1.9}$ | $61.1_{2.2}$ |
| | PBFT | $62.8_{1.3}$ | $68.1_{1.4}$ | $65.9_{1.8}$ | $61.3_{1.2}$ |
| | WILDA-S | $64.8_{0.4}$ | $70.3_{0.6}$ | $67.8_{2.1}$ | $65.0_{1.1}$ |

using the following template: "Demonstration: {*demonstration*}. Answer: ({*answer*})." The adapter learned to encode demonstration-specific information indirectly by aligning its outputs with the teacher's responses, without explicitly seeing the demonstration itself. After training, the similarity between the student model's response and the original demonstration was computed. Table 12 shows the average BERTScore similarity [43] between the original demonstrations and the student's reconstructed response.

The consistently high BERTScore values across all datasets indicate that the student model can reliably retrieve the encoded demonstration from the adapter. This suggests that WILDA effectively disentangles and stores task-specific information within the adapter's weights. Notably, when compared to standard ICL, WILDA often produced different outputs for certain queries, particularly in instances where it corrected "corrupted" labels provided by the teacher. Despite these differences, the student model maintained a high degree of semantic similarity in reproducing the demonstrations. This suggests that the adapter weights capture not only the demonstration itself but also additional latent information that contributes to improved generalization.

We present below a pair of examples from SST and RTE, chosen to represent reconstructed demonstrations with similarity scores close to the dataset averages.

Table 11: Cross-domain impact of WILDA. To test whether task-specific adaptation harms general performance, we pair *unrelated* GLUE datasets and evaluate whether demonstrations from one domain deteriorate performance on another. Each model variant is exposed to 16 demonstrations from the unrelated domain. **Demo** denotes the dataset used for constructing demonstrations (i.e., the adapter's training domain), while **Eval** indicates the unseen evaluation task. We report mean accuracy across 10 seeds. WILDA-S refers to the model with the task-specific adapter *enabled* at inference.

| Demo | Eval | 0-shot | 16-shot | WILDA-S |
|---|---|---|---|---|
| QNLI | RTE | 62.3 | $61.9_{1.2}$ | $62.1_{0.9}$ |
| SST | COLA | 44.6 | $44.9_{0.5}$ | $44.7_{0.4}$ |
| MNLI | QQP | 61.1 | $60.9_{0.6}$ | $61.0_{0.7}$ |
| MRPC | SST | 79.1 | $81.4_{1.4}$ | $81.3_{1.1}$ |

Table 12: Average BERTScore ($F_1$) similarity across GLUE datasets. Higher scores indicate better fidelity in recalling the encoded demonstration.

|            | RTE  | SST  | QNLI | MNLI | COLA | MRPC | QQP  |
|------------|------|------|------|------|------|------|------|
| BERTScore  | 0.84 | 0.91 | 0.80 | 0.83 | 0.86 | 0.82 | 0.81 |

---

**SST: Example 1**

**Original:** *Proves once again he hasn't lost his touch, delivering a superb performance in an admittedly middling film.*
**Answer:** (Positive)

**Reconstructed:** *He demonstrates once more that he hasn't missed a beat, delivering a remarkable performance in what is admittedly an average film.*
**Answer:** (Positive)

---

**SST: Example 2**

**Original:** *Though many of the actors spark briefly when they first appear, they can't generate enough heat in this cold vacuum of a comedy to ignite a reaction.*
**Answer:** (Negative)

**Reconstructed:** *Although some actors manage to show a hint of energy early on, they fail to create any real warmth or spark within this lifeless and chilly comedy.*
**Answer:** (Negative)

---

**RTE: Example 1**

**Original:**
Premise: *The source added that the investigation proved that the bases of the genocide crime "were completed with a series of illegal arrests followed in some cases with assassinations or cases of disappearances and were preceded, according to information attached to the file, by cases of torture."*
Hypothesis: *Investigators discovered that a series of illicit arrests were often followed by disappearances or murders and were preceded by torture.*
**Answer:** (True)

**Reconstructed:**
Premise: *The investigation confirmed that genocide involved illegal arrests followed by disappearances or murders, often preceded by torture.*
Hypothesis: *Investigators found that unlawful arrests frequently resulted in disappearances or murders, often preceded by acts of torture.*
**Answer:** (True)

---

**RTE: Example 2**

**Original:**
Premise: *American tobacco companies were showing a profit most quarters due to export sales of cigarettes and diversification of products sold, including food.*
Hypothesis: *PM often entered markets with both cigarettes and food.*
**Answer:** (False)

**Reconstructed:**
Premise: *Profitability was often maintained by American tobacco companies through diversi-*

Table 13: Summary of the models used in the experiments, including their Hugging Face IDs, parameter counts, context window sizes, training token volumes, and adapter sizes.

| Model | Hugging Face ID | Parameters | Context window size | Training tokens | Adapter size |
|---|---|---|---|---|---|
| Llama 3 | Meta-Llama-3-8B | 8B | 8k | 15T | 21M |
| Llama 2 | Llama-2-7b | 7B | 4k | 2T | 20M |
| Phi 3 | Phi-3-mini-4k-instruct | 3.8B | 4k | 3.3T | 4.5M |

*fication into food products and successful cigarette exports.*
Hypothesis: *Philip Morris International offered food items and cigarettes.*
**Answer:** (False)

# E  Experimental Details

## E.1  Models

For all three models – Llama 3, Llama 2, and Phi 3 – we utilize the `bfloat16` half-precision format for parameters. A summary of the models is provided in Table 13.

## E.2  Hyperparameters

We employ the AdamW optimizer [29] for both PBFT and WILDA variants, with a learning rate of $10^{-4}$. For ICV [28] and Batch-ICL [42], we follow the implementations provided in the original papers and adapt them to our codebase, using their default parameters where specified. In the case of Batch-ICL, we utilize attention heads from the last 20 layers ($k = 20$) and fine-tune the model for 10 epochs.

**LoRA adapter configuration.**

- **Rank** ($r = 8$): Dimensionality of the low-rank matrices used to approximate the original weights, controlling parameter efficiency.

- **Scaling factor** ($\alpha = 32$): Multiplier that scales the low-rank updates relative to the frozen base weights.

- **Dropout**: (0.1): Regularization applied to the low-rank updates during training.

- **Target modules**: `q_proj`, `k_proj`, `v_proj`, `o_proj`, `gate_proj`, `up_proj`, `down_proj`.

## E.3  Computing Infrastructure

We conducted our experiments on *AMD Ryzen Threadripper 3970X 32-Core Processors* and $4\times$ *NVIDIA GeForce RTX 3090* GPUs with 24GB of RAM.

# F Prompt Templates

## F.1 GLUE Prompt Structure

---

**Generic prompt template for GLUE tasks**

**Demonstrations:**
```
{Sentence 1}
{Sentence 2 (if applicable)}
Answer: ({Correct answer})
```
**Query:**
```
{Sentence 1}
{Sentence 2 (if applicable)}
Question: {Task-specific question}
Answer: (
```

---

The prompts for GLUE tasks typically consist of two sentences (or one in certain cases) followed by a task-specific question and the corresponding answer. The model is expected to choose from predefined labels like *Yes/No*, *True/False*, or specific class names based on the dataset. The phrasing of the question preceding each answer in the demonstrations is specific to the task. Below is a list of the questions used for each GLUE dataset. To encourage the model to select from predefined labels, we prepend the phrase "answer with one word" before each question, and we append clarifying options such as *Yes or No?* to prompt a more targeted response:

- **RTE:** {hypothesis} True or False?
- **SST:** What is the sentiment? Positive or Negative?
- **QNLI:** Does the sentence answer the question? Yes or No?
- **MNLI:** Is the second sentence an Entailment, Contradiction, or Neutral?
- **CoLA:** Is this sentence linguistically acceptable? Yes or No?
- **MRPC:** Do both sentences say the same thing? Yes or No?
- **QQP:** Do both questions ask the same thing? Yes or No?

## F.2 MMLU prompt structure

---

**Generic prompt template for MMLU sub-datasets**

**Demonstrations:**
```
Question: {Previous Question 1}
Answer choices:
 (A: {Choice A1}),
 (B: {Choice B1}),
 (C: {Choice C1}),
 (D: {Choice D1})
Answer: (Correct Answer 1)

Question: {Previous Question 2}
Answer choices:
(A: {Choice A2}),
(B: {Choice B2}),
(C: {Choice C2}),
(D: {Choice D2})
Answer: (Correct Answer 2)
...
```

---

**Query:**

```
Question: {Current Question}
Answer choices:
(A: {Choice A}),
(B: {Choice B}),
(C: {Choice C}),
(D: {Choice D})
Answer: (
```

**Example for MMLU `elementary_math` (MATH)**

**Demonstrations:**

```
Question: Ms. Perez drove a total of 40 miles in 5 days.
She drove the same number of miles each day.
How many miles did Ms. Perez drive each day?
Answer choices: (A: 5), (B: 7), (C: 8), (D: 9)
Answer: (C: 8)

Question: Find the median in the set of data
23, 13, 18, 29, 32, 25.
Answer choices: (A: 18), (B: 24), (C: 25), (D: 29)
Answer: (B: 24)
```

**Query:**

```
Q: A worker on an assembly line takes 7 hours to produce
22 parts. At that rate how many parts can she produce
in 35 hours?
Answer choices:
(A: 220 parts),
(B: 770 parts),
(C: 4 parts),
(D: 110 parts)
Answer: (
```

