# OpenReview forum: "Disentangling Latent Shifts of In-Context Learning with Weak Supervision"
_NeurIPS.cc/2025/Conference — NeurIPS 2025 poster_

### Official Review · Reviewer_xixy · 2025-07-01

**Clarity:** 3
**Significance:** 3
**Originality:** 3
**Rating:** 4
**Confidence:** 4

**Summary:**

In-context learning (ICL) allows LLM to do few - shot learning using labeled prompt examples but faces instability as prompts grow longer. To tackle this, a parameter - efficient method is proposed. It treats ICL as weak supervision, disentangling latent shifts from demonstrations and queries. An ICL - based teacher creates pseudo - labels for unlabeled queries. A student, updating a lightweight adapter, predicts these labels using only query inputs, capturing demonstration effects compactly for efficient inference and compatibility with new demos. Trained on noisy teacher outputs, the student often outperforms the teacher via pseudo - label correction and coverage expansion, following the weak - to - strong generalization. Empirically, the method boosts generalization, stability, and efficiency on in - domain and out - of - domain tasks, outperforming standard ICL and prior disentanglement approaches.

**Questions:**

1.Based on my understanding, this method can compress the information from the training set demonstrations into the adapter. If there is a new test sample with demonstrations that are significantly different from the existing training set (for example, the existing demonstration training set is about mathematics, while the new one is about physics), and the adapter does not store the corresponding physics knowledge, then the given demonstrations cannot be utilized. In this case, is this method still meaningful?

**Ethical Concerns:**

["NO or VERY MINOR ethics concerns only"]

**Final Justification:**

In my opinion, in-context learning—which enables learning a new task from context—possesses task-level generalization ability. However, this LoRA adapter cannot adaptively adjust its parameters for new tasks based on new task-specific context and thus lacks such task-level generalization ability.

Overall, the proposed model may not be fair compared with original in-context learning.

Upon further consideration, I do not believe this constitutes an effective improvement for in-context learning; instead, it merely learns an effective LoRA adapter. Consequently, the significance of this paper would be diminished.

**Limitations:**

see question

**Quality:**

3

**Strengths And Weaknesses:**

Strengths:

1. The paper presents a novel and effective approach to disentangle latent shifts in in-context learning, leveraging a weak-to-strong framework that demonstrates clear innovation and practical efficacy.

2. The manuscript is well-structured, with logical flow and concise expression, making it accessible and easy to follow for readers.


3. The research motivation is well-articulated, rooted in addressing the core challenge of ICL instability, and the proposed technical route is rigorously designed with feasible implementation steps. The basic assumption is that the teacher can get robust result, which is similar to the W2S framework.

4. The extensive experimental setup, covering diverse in-domain and out-of-domain tasks, provides robust evidence to validate the model's effectiveness, including improvements in generalization, stability, and efficiency.


Weaknesses:

1. The analysis of why the student model can outperform the teacher model through pseudo-label correction and coverage expansion lacks sufficient theoretical depth, and the specific mechanisms behind the "weak-to-strong generalization effect" in the context of in-context learning need to be further clarified.

---

> ### Author Rebuttal · Authors · 2025-07-30
>
> Thank you for your thoughtful comments and for noting both the strengths and areas where clarification is needed. Below, we offer two complementary perspectives in response to your question about unseen domains, and we provide additional context on our theoretical analysis and empirical results.
>
> ### Perspective 1: Preservation of general capabilities and flexible few‑shot use
> Like other parameter‑efficient fine‑tuning methods, WILDA’s adapters add only 0.1–0.3 % of the base model’s parameters. This design means that the underlying language model remains intact: during training we toggle the adapter on and off, and at inference time users can disable it to recover the original zero‑shot behavior.
>
> Prompted by your comment and a similar observation from another reviewer, we conducted additional experiments with Llama 3 (8B) to verify that activating an adapter trained on one domain does not hinder zero‑shot performance on unrelated tasks. Using non‑overlapping GLUE dataset pairs, we observed that performance differences remain within one standard deviation (10 different seeds), confirming that WILDA preserves general zero‑shot capabilities on unrelated queries. The method thus provides a modular, task‑specific memory of demonstration effects that can be activated when relevant and safely bypassed otherwise. The results are as follows:
> | Demonstration dataset | Evaluation dataset (unseen task) | 0‑shot baseline without unrelated demonstrations | 16‑shot ICL | WILDA‑S (16-shot adapter on, 0‑shot prompt) |
> |-|-|-|-|-|
> | QNLI | RTE | 62.3 | 61.9 (1.2) | 62.1 (0.9) |
> | SST | COLA | 44.6 | 44.9 (0.5) | 44.7 (0.4) |
> | MNLI | QQP | 61.1 | 60.9 (0.6) | 61.0 (0.7) |
> | MPRC | SST | 79.1 | 81.4 (1.4) | 81.3 (1.1) |
>
> Moreover, WILDA does not preclude standard ICL. Once the adapter is trained, we can still supply new demonstrations in the prompt and combine them with the adapter’s latent shift. Appendix D.2 (Table 9) reports results where we encode 16 demonstrations in the adapter and then provide another 16 demonstrations in the prompt at inference. This WILDA‑S (16/16) configuration outperforms the standard 32‑shot ICL across all evaluated datasets. In other words, the student can benefit from both the encoded shift and additional demonstrations.
>
> ### Perspective 2: Domain relevance and the need for task‑specific adapters
> Your hypothetical scenario – training an adapter on math demonstrations and then providing physics demonstrations at test time – highlights an important limitation of any ICL method: the usefulness of demonstrations depends on their relevance to the query. In standard ICL, if you mix math and physics demonstrations, the model will mostly attend to the physics examples when answering a physics question, because those tokens more closely match the query. Similarly, a WILDA adapter trained on math will encode how the model’s activations should shift when it sees math‑related queries. Applying that adapter to a physics query does not magically inject physics knowledge; the shift is largely ignored by the model because it does not align with the query’s content. To handle physics problems, one should train a separate physics adapter (on unlabeled physics queries with corresponding demonstrations) and switch it on for physics questions. Importantly, this does not diminish the utility of WILDA for math tasks; you are free to toggle the math adapter on for math queries or turn it off entirely. This also relates to the other reviewer’s question about the impact on the base model’s general capabilities: the model effectively ignores the adapter’s contribution when the query domain does not align with the adapter’s training domain.  Thus, the method remains meaningful: it provides a parametric memory of demonstration effects that can be activated when relevant and bypassed when not.
>
> ### On the weak‑to‑strong analysis
> We agree that a deeper theoretical connection between weak-to-strong generalization (W2S) and in-context learning would be valuable. Our paper provides an empirical analysis of W2S phenomena and draws parallels with known theory. Specifically, we approximate the Lipschitz constant of the model by the Frobenius norm of its input–output Jacobian and find that WILDA exhibits a significantly lower Lipschitz constant than standard ICL or PBFT. A low Lipschitz constant implies that the model’s predictions vary smoothly with small changes in input, a key ingredient for correcting noisy pseudo‑labels. We also track the rate of pseudo‑label correction over training epochs and show that it increases steadily, indicating that the student is refining the teacher’s labels rather than merely copying them. Finally, we discuss coverage expansion, where local corrections propagate to nearby inputs in representation space, gradually broadening the region where the model makes confident predictions. These empirical observations align with the W2S framework: local consistency facilitates pseudo‑label correction, which in turn leads to broader generalization. Formalizing this link between W2S and ICL remains an exciting direction for future work.

---

> > ### Comment · Reviewer_xixy · 2025-08-04
> > **Concern on Domain relevance and the need for task‑specific adapters**
> >
> > In my opinion, in-context learning— which enables learning a new task from context—possesses task-level generalization ability. However, your LoRA adapter cannot adaptively adjust its parameters for new tasks based on new task-specific context, and thus lacks such task-level generalization ability.

---

> > > ### Author Response · Authors · 2025-08-04
> > >
> > > Thank you for your comment. We fully agree with your observation: in the current WILDA framework, each distinct set of demonstrations requires training a new lightweight adapter through fine-tuning. This limitation is shared by most parameter-efficient tuning approaches, which similarly do not adapt parameters for unseen tasks without additional optimization.
> > >
> > > Unlike standard in-context learning, which performs transient adaptation solely through the prompt, WILDA explicitly encodes the effect of demonstrations into a lightweight adapter. This introduces a trade-off: while standard prompting can handle new tasks immediately, it suffers from instability and increased computational cost as the prompt grows longer. WILDA instead performs a one-time fine-tuning of a small adapter, resulting in a persistent, reusable parametric shift that mitigates long-context instability and reduces inference cost when demonstrations recur.
> > >
> > > As a potential future direction, we are exploring adapter generators, networks capable of producing adapter parameters directly from raw input text. Once trained, such a generator could quickly synthesize task-specific adapters without requiring further fine-tuning, enabling a more adaptive, task-level generalization aligned with the spirit of in-context learning.

---

> > > > ### Comment · Reviewer_xixy · 2025-08-04
> > > > **Thanks for your further clarification.**
> > > >
> > > > But in my opinion, the proposed Lora adapter has to face the long-context challenge, the same as in-context learning, due to relying on context.
> > > >
> > > > Overall, the proposed model may not be fair compared with original in-context learning.

---

> > > > > ### Author Response · Authors · 2025-08-04
> > > > >
> > > > > ### Long context
> > > > >
> > > > > We appreciate your continued engagement. We respectfully disagree with the assertion that our method faces the same long-context challenge as standard in-context learning. Our method was specifically designed to mitigate this limitation, and we provide empirical evidence to support this claim.
> > > > >
> > > > > 1. Unlike standard ICL, which must fit all demonstrations into a single prompt, WILDA can handle large demonstration sets by splitting them into subsets. For each subset, we fine-tune a lightweight adapter and then merge these adapters by summing their parameters, as detailed in Section 3.3 (Adapter Arithmetic). This approach allows us to scale, potentially even beyond the model’s context window, while improving performance. After merging, we obtain a single adapter of the same size as one trained on a single subset, so there is no growth in inference-time parameters or memory cost.
> > > > > 2. Long prompts in standard ICL are known to cause unstable behavior. WILDA mitigates this issue by parameterizing the latent shift induced by demonstrations in a compact adapter, which, as shown in Table 2, produces significantly more stable predictions than ICL as well as stronger performance.
> > > > >
> > > > >
> > > > > ### Method comparison
> > > > >
> > > > > We believe it is important to compare against standard ICL not because WILDA is a one-to-one replacement, but because ICL serves as the teacher in our weak-to-strong framework and represents the natural baseline for few-shot adaptation.
> > > > >
> > > > > Our results show that the student adapter consistently outperforms the ICL teacher, demonstrating that WILDA captures and improves upon the underlying weakly supervised signal. In addition to this teacher–student comparison, we conduct an extensive evaluation against other approaches that require fine-tuning, including Pattern-Based Fine-Tuning with gold labels and BatchICL.

---

### Official Review · Reviewer_A5UG · 2025-07-02

**Clarity:** 3
**Significance:** 2
**Originality:** 2
**Rating:** 4
**Confidence:** 4

**Summary:**

The paper investigates a possible way to internalize in-context learning (ICL) capability, i.e., learning a set of adaptors. The authors first illustrate on why disentangling latent shift is an (approximately) effective way of inducing ICL, and then propose a teacher-student learning framework to acquire the latent shift while encompassing also tje feed-foward layers and activation functions. Empirically, the proposed method demonstrate good gains for in-domain data as well as out-of-domain data under the same task definition, outperforming other baseline methods by a clear margin.

**Questions:**

There is a recent work "Implicit In-context Learning" also concerns a very similar setting. What's the relation and performance comparing to it?

**Ethical Concerns:**

["NO or VERY MINOR ethics concerns only"]

**Final Justification:**

During the rebuttal, my concerns regarding the computational cost and generalization ability of learned adapters are adequately  addressed. Going over other reviewer's feedback and discussion, I decided to maintain my score and vote towards acception.

**Limitations:**

Yes

**Quality:**

3

**Strengths And Weaknesses:**

**Strengths**:
1. The motivation and presentation is logically sound and clear.
2. The proposed method is simple yet very effective for in-task learning.
3. The empirical design is reasonable and convincing.

**Weakness**:
1. The internalization process (i.e., learning the lora) is to some extent heavy, hindering one the mostly appreciated feature of ICL, ie, flexibility. This lead to a concrete question of the reviewer, how many extra computes (e.g., FLOPs) will the process need to learn the lora, and how does this computational cost comparing to other baseline methods.
2. The effectiveness of the proposed method essentially constrained by the task definition. The learned adaptor will not be able to generalize beyond the learned task.
3. While adding a task adaptor to the backbone model, will it hinder the general capability of the model? There might be regression on other tasks.

---

> ### Author Rebuttal · Authors · 2025-07-30
>
> Thank you for your thoughtful review. We appreciate your recognition of our work’s clarity and empirical effectiveness, and we address your points below.
>
> ### Computational cost of learning the adapter
> We share your view that one of the major appeals of in‑context learning is its flexibility. WILDA was designed to minimize overhead while maintaining this flexibility. The adapter we learn uses LoRA modules that constitute only 0.1–0.3 % of the base model parameters: for Llama 3 (8B), this corresponds to ~21 M parameters, and only ~4.5 M for Phi‑3. Because the base model is frozen, only these additional weights are updated.
>
> To quantify compute, we measure training cost relative to the cost of processing a 16‑shot prompt on the same backbone. For Llama 3, a 16‑shot prompt takes roughly 120x longer than a zero‑shot query because attention cost grows quadratically with prompt length. Fine‑tuning the adapter for ten epochs on 100 unlabeled queries requires about 2100 forward passes of a 16‑shot prompt. In other words, the training cost is equivalent to answering around two thousand 16‑shot queries. Once the adapter is trained, we no longer feed demonstrations at inference, so each subsequent query runs at zero‑shot speed. For workloads serving more than a few thousand queries, the one‑off cost is amortised, and the total compute is lower than repeatedly processing long prompts.
>
> Moreover, long prompts often exceed the model’s context window or lead to unstable performance; to address this, WILDA supports partitioning the demonstration set across multiple adapters. Each adapter is trained on a manageable subset of demonstrations, and these adapters can later be fused by parameter addition. This strategy not only mitigates context‑length limitations and reduces memory overhead but also preserves stability: instead of processing a single long, order‑sensitive prompt, we store several compact “latent shift” adapters and combine them as needed, fitting within the available context window and ensuring consistent performance.
>
> ### Task specificity and generalization
> An adapter is by design intended to specialize the model toward a given task, and in WILDA, this is achieved by encoding the latent shift induced by task‑specific demonstrations. Our experiments suggest that this specialization is both effective and reasonably robust: across seven GLUE tasks and two MMLU subsets, WILDA‑S improves over standard ICL by 2.6–11.9 points, and on out‑of‑domain task pairs (e.g., training on QNLI and testing on RTE), it maintains higher accuracy than baseline methods. This indicates that the adapter likely captures a task‑level representation rather than memorizing individual examples. Additionally, our “adapter arithmetic” experiments show that multiple adapters trained on different demonstration subsets can be fused, allowing the model to combine information from larger or more varied prompts without exceeding the context window. We acknowledge that transfer across entirely different tasks remains limited, and exploring multi‑task adapters or hierarchical parameterizations is an interesting direction for future work.
>
> Impact on the base model’s general capabilities
> LoRA adapters are implemented as additive residuals and do not modify the base model’s weights. During training, we toggle the adapter on (student) and off (teacher), and at inference, it can be enabled or disabled without altering the core model. Importantly, additional experiments (see table below) we conducted in response to your remark show no regression in zero‑shot performance, even when an adapter trained on one domain remains active during evaluation on a completely different domain (a scenario conceivable in practice, where the model may be presented with ad hoc queries) – accuracy differences remain within normal variability.
>
> To quantify this effect, we evaluated Llama‑3 (8B) on unrelated, non‑overlapping dataset pairs from GLUE, where both 16‑shot ICL and WILDA‑S are exposed to task‑specific demonstrations from a different domain.  Results show that performance differences remain well within one standard deviation (averaged over 10 seeds), demonstrating that WILDA‑S does not harm general zero‑shot capabilities. We will include these results in the revised manuscript.
> | Demonstration dataset | Evaluation dataset (unseen task) | 0‑shot baseline without unrelated demonstrations | 16‑shot ICL | WILDA‑S (16-shot adapter on, 0‑shot prompt) |
> |-|-|-|-|-|
> | QNLI | RTE | 62.3 | 61.9 (1.2) | 62.1 (0.9) |
> | SST | COLA | 44.6 | 44.9 (0.5) | 44.7 (0.4) |
> | MNLI | QQP | 61.1 | 60.9 (0.6) | 61.0 (0.7) |
> | MPRC | SST | 79.1 | 81.4 (1.4) | 81.3 (1.1) |
>
>
> ### Relation to “Implicit In‑context Learning”
> The recently published Implicit In‑Context Learning (I2CL) paper (Li et al., 2025) proposes condensing all demonstration examples into a single context vector. Specifically, each demonstration is passed through the base model to produce a demonstration vector; these vectors are then aggregated in a permutation‑invariant manner (e.g., via averaging and a small MLP) to form the context vector. At inference, this context vector is linearly combined with the query’s activations and injected into each residual block of the model. The combination weights (“calibration coefficients”) are learned for each layer using a small calibration set. This design eliminates the need to prepend demonstration tokens, while also reducing memory usage and speeding up inference. However, a closer look at their results reveals that I2CL regularly underperforms compared to standard ICL, indicating a trade‑off between efficiency and accuracy.
>
> WILDA‑S, in contrast, uses a teacher–student framework: a teacher model generates pseudo‑labels from few‑shot prompts, and a student fine‑tunes a small adapter to mimic and improve upon those outputs. This mechanism captures the non-linear latent shift induced by demonstrations, extending beyond the attention layers to include feed-forward networks and activation functions, and enables weak-to-strong generalization, where the student corrects noisy pseudo-labels and expands coverage. Empirically, WILDA‑S consistently surpasses standard ICL and enables handling long context by splitting the demonstrations into multiple subsets and then merging them through adapter arithmetic. Furthermore, the student can still benefit from new demonstrations supplied in the prompt: when we encode 16 demonstrations into the adapter and provide 16 more in the prompt, WILDA‑S (16/16) outperforms a 32‑shot ICL baseline across all datasets (Table 9 in Appendix D.2).

---

### Official Review · Reviewer_o4hd · 2025-07-02

**Clarity:** 4
**Significance:** 3
**Originality:** 3
**Rating:** 5
**Confidence:** 5

**Summary:**

This paper proposes to use a lightweight adapter to learn the latent shifts of ICL demonstrations. By learning from the teacher model which is given the demonstrations, the student model learns to perform similarly under zero-shot setting.

**Questions:**

Please see the weakness.

**Ethical Concerns:**

["NO or VERY MINOR ethics concerns only"]

**Final Justification:**

The authors have addressed all my concerns. The method is simple yet effective. One of the great contributions of this work is that it provides a solution to the example order problem in ICL. Therefore, I raise my rating.

**Limitations:**

Yes

**Quality:**

3

**Strengths And Weaknesses:**

**Strengths**:
* This paper avoids too much simplification and the uncontrollable bias that previous methods have.
* The proposed **WILDA-S (Shuffle)** is a clever way to overcome the instability of demonstration orders, which standard ICL suffers from.
* It shows superiority over the baseline methods.

**Weaknesses**:
* The unavoidable training process is the biggest weakness of this work. In terms of computation resources, this method requires more GPU memory and computation, prohibiting some scenarios, like in edge devices. In terms of time consumption, it's terrible if one always has to fine-tune an adapter before using ICL.
* It would be better to present the average scores on these datasets.
* GLUE and MMLU are relatively old and easy datasets. I want to see the performance on harder datasets, e.g., Big-bench-hard, AGIEval, ARC-challenge.

---

> ### Author Rebuttal · Authors · 2025-07-30
>
> Thank you for your constructive feedback and for noting the strengths of our approach. We appreciate your recognition of WILDA’s ability to mitigate order sensitivity and outperform baseline methods. Below, we respond to your concerns and clarify the motivations behind our design choices.
>
> ### Computational overhead and the “investment” perspective
> We acknowledge that WILDA introduces a lightweight fine‑tuning step and understand the reviewer’s concern about compute cost. However, we believe the initial compute cost should be viewed as an investment rather than a recurring cost. The adapter is extremely small; LoRA modules constitute only 0.1–0.3% of the base model parameters, which makes the process computationally lightweight. Once trained, the adapter can be toggled on or off at inference. As our limitations section notes, processing a 16-shot prompt on Llama‑3 is about 120× slower than zero-shot because the self-attention cost grows quadratically with prompt length. Fine-tuning the adapter with 100 unlabeled queries costs the equivalent of ~2,100 such 16-shot inference runs. Thus, in scenarios where one expects to answer more than a few thousand queries, the one‑time training cost gets amortised, and the overall wall‑clock time is reduced because the model no longer needs to re‑process demonstrations for every query.
>
> In addition to amortising the one‑time training cost, partitioning demonstrations across multiple adapters also mitigates context‑window limitations. As Table 4 and the accompanying discussion explain, each adapter is trained on a manageable subset of demonstrations; merging them via adapter arithmetic allows us to incorporate many examples without exceeding the model’s context length. This strategy not only reduces memory overhead but also avoids the quadratic scaling of self‑attention for long prompts. By distributing the prompt across several small adapters, we fit within a single GPU and lower inference cost. At inference, we load only the necessary adapters and avoid the quadratic cost of long prompts, which can be beneficial even for edge devices. We acknowledge that WILDA is less suitable when one has abundant labeled demonstrations and must process new demonstration sets for every query, where we need to fine-tune an adapter for each new prompt and may negate the amortization advantage. In many real‑world applications, however, labeled demonstrations are scarce, and the same few examples are reused across many queries.
>
> Moreover, long demonstration sequences themselves can lead to unstable predictions. Standard ICL is highly sensitive to the order and selection of demonstrations, and performance often degrades as prompts grow longer. Our experiments show that WILDA’s parametric approach, encoding the latent shift into a fixed adapter, significantly improves stability: across multiple random seeds, WILDA‑S exhibits markedly lower variance than standard ICL. This stability stems from learning a reusable representation of the task rather than relying on the precise ordering of a long prompt. Thus, WILDA not only alleviates context‑length constraints but also delivers more consistent behavior.
>
> These additional points strengthen the view of WILDA as an upfront investment: by training a small adapter once and partitioning demonstrations when necessary, we both reduce inference cost and improve stability, making the method practical even when context windows are limited or prompts are long.
>
> ### Presentation of results and “average scores”
> We report results as means over multiple runs (with different random seeds), along with standard deviations, for each dataset. For example, Table 1 includes averages over ten runs. If the reviewer is referring to averaging scores across different tasks into a single number, we deliberately avoid this because it may obscure task‑specific behavior and can be misleading when tasks have different scales or importance. We are therefore unsure what kind of average the reviewer would find helpful. We welcome further clarification and will be happy to include additional aggregates in the final version if that aligns with community standards.
>
> ### Evaluation and harder datasets
> We chose GLUE and selected MMLU subsets because they are well‑established NLP benchmarks that allow fair comparison with prior work. In this work, we focused on models under 10 billion parameters due to computational constraints; such models are typically better aligned with GLUE/MMLU‑style tasks than with extremely challenging reasoning benchmarks. Importantly, WILDA is task‑agnostic, and there is nothing in the approach that fundamentally limits its application to harder datasets.
>
> Nonetheless, prompted by your remark, we evaluated WILDA on the ARC-Challenge dataset. Using Llama‑3 (8B) across all methods, we observed that WILDA-S achieves the highest accuracy with improved stability, outperforming the baselines and related approaches (see table below). This trend is consistent with our results on GLUE and MMLU. We will include these findings and the corresponding table in the revised manuscript.
> | Method | Accuracy (ARC-challenge) | Std. dev. |
> |-|-|-|
> | Zero-shot     | 38.3  | - |
> | 16-shot ICL   | 54.5  | 1.8 |
> | PBFT  | 50.8  | 1.5 |
> | ICV | 49.9  | 1.7 |
> | Batch-ICL  | 48.6 | 2.0 |
> | WILDA-F  | 55.7 | 1.3 |
> | WILDA-S  | **59.2**  | **1.0** |
> | WILDA-R  | 56.5 | 1.5 |

---

> > ### Comment · Reviewer_o4hd · 2025-08-06
> >
> > I sincerely thank the authors for the clarification.
> >
> > Regarding the "average scores", I refer to the average over different tasks. As prior works usually average the scores over different subtasks to represent the holistic performance on GLUE or MMLU, I suggest the authors report this as well, at least in the appendix.
> >
> > The improvement on ARC-challenge dataset looks great. Thank you!

---

> > > ### Author Response · Authors · 2025-08-07
> > >
> > > Thank you for the positive follow-up and for clarifying your suggestion regarding average scores. We appreciate your feedback and will include the task-averaged results in the final version of our paper.

---

> > > > ### Comment · Reviewer_o4hd · 2025-08-07
> > > >
> > > > I don't have further questions and have raised my rating.

---

> > > > > ### Author Response · Authors · 2025-08-07
> > > > >
> > > > > Thank you, we appreciate it!

---

> ### Author Response · Authors · 2025-08-06
> **Gentle reminder -- rebuttal review**
>
> Just a quick follow-up to kindly check if you’ve had a chance to review our rebuttal. We’d be happy to provide any additional clarification if needed and hope our response addressed your concerns.

---

### Official Review · Reviewer_ECRJ · 2025-07-03

**Clarity:** 3
**Significance:** 1
**Originality:** 1
**Rating:** 3
**Confidence:** 5

**Summary:**

In-context learning (ICL) enables large language models to adapt to new tasks without updating model weights. However, the choice and ordering of demonstrations in the prompt often lead to unstable predictions and poor generalization. While prior work has explored how latent representations shift during ICL—typically by approximating internal mechanisms—this paper takes a more direct approach. The authors treat the outputs of ICL as weak supervision signals and distill the predictions from few-shot demonstrations into a student model. This approach captures both the effects of demonstrations and the underlying latent shifts. Empirically, their proposed framework, WILDA, outperforms pure prompting strategies and prior methods that attempt to modify internal representations, achieving better results on both in-domain and out-of-domain tasks.

**Questions:**

* The student model uses the same architecture and size as the teacher, which is unusual. Most distillation setups use a smaller or simpler student model to encourage compression or efficiency. Can the author clarify this?
* Why are MMLU subjects split into math and others, especially elementary math, is the dataset targeted to "elementary math"?
* A useful experimental setup to better study the demonstration effect would be as follows:
(1) In an 8-shot demonstration, use one high-quality (gold) example and seven dummy examples.
(2) Compare this to a 1-shot prompt using only the gold example.
(3) This could help answer whether the student model has actually learned/captured to denoise bad demonstrations and upweight the important ones, reflecting a true understanding of the latent shift rather than just copying the teacher’s outputs.

**Ethical Concerns:**

["NO or VERY MINOR ethics concerns only"]

**Final Justification:**

I have put my final concern of this paper in the comment.

**Limitations:**

Yes, they discussed in Appendix C.

**Paper Formatting Concerns:**

Yes. They follow the format.

**Quality:**

2

**Strengths And Weaknesses:**

[Strengths]
* Clarity: The paper is well-written and easy to follow.

[Weaknesses]
* The biggest concern is that **the teacher-student framework mostly looks like a standard distillation process**, where the student learns from the teacher’s pseudo-labels. It’s unclear whether the student model is actually learning the latent shift caused by the few-shot demonstrations, or just learning to copy the teacher’s predictions. It’s also not clear if the student is learning how to properly aggregate these weak supervision signals. Moreover, since the student model never directly sees the demonstrations, it might simply be mimicking the teacher’s outputs, without truly learning the effects of the demonstrations.
* For the baseline comparison, how does WILDA compare to a zero-shot distillation baseline? For example, distilling from a zero-shot-prompted teacher instead of few-shot. Is the performance gain significant (i expect we will see improvements but how large they are)? A follow-up question is that how does the current distilled student perform when it is prompted with few-shot examples again? What's the performance gap between the student’s zero-shot and few-shot scenarios? These two are different, one is comparing a student model distilled from few-shot prompting and zero-shot prompting; the other one is a few-shot distilled student model making inference using zero-shot and few-shot prompt.
* About Table 4: The performance improvements might just come from merging multiple adapters, each of which already does better than ICV and Batch-ICL on their own. So it’s unclear whether the gains really come from modeling the latent shift of demonstrations, or if it’s just combining stronger components.

---

> ### Author Rebuttal · Authors · 2025-07-30
>
> We thank the reviewer for the feedback and appreciate the opportunity to clarify our contributions further and address the concerns raised.
>
> ### WILDA and distillation
> While distillation is a useful tool and WILDA’s teacher–student setup resembles it, our goal is fundamentally different. Rather than compressing a large model into a smaller one, WILDA focuses on capturing the latent shift induced by demonstrations, which can mitigate the problems with long context down the line. In our setup, the teacher processes the full demonstration–query prompt and outputs a probability distribution, while the student, equipped with a small LoRA adapter (0.1–0.3% of parameters), sees only the query and learns to approximate the teacher’s distribution. By keeping teacher and student architectures identical, we isolate the effect of the adapter and show that gains come from encoding latent shifts, not differences in model capacity.
> This approach is orthogonal to model compression: in principle, one could use a smaller student model while still leveraging WILDA to capture demonstration‑induced shifts. Importantly, matching the teacher’s outputs is a necessary step for the student to internalize these shifts, but WILDA goes beyond simple imitation. In fact, the student often surpasses its teacher, with WILDA‑S improving standard ICL by 2.6–11.9 points on GLUE and MMLU for Llama‑3. This behavior reflects weak‑to‑strong (W2S) generalization: during training, WILDA corrects noisy pseudo‑labels, expands coverage, and achieves lower Lipschitz constants (Section 4).
>
> To further demonstrate that the adapter encodes demonstration content rather than merely copying the teacher’s outputs, we conducted an experiment (Appendix D.3) treating the adapter‑induced latent shift as a “task vector.” Retrieving tokens via nearest‑neighbor search over the vocabulary produces text that closely matches the original demonstrations. This aligns with the intuition that, because the student only observes the query while the teacher has access to both demonstrations and the query, it cannot reach teacher‑level performance unless it internally represents the missing demonstration information. Our empirical results confirm this: zero-shot performance is consistently worse than few-shot teacher predictions, and the student closes this gap, often surpassing the teacher, by encoding and leveraging these demonstration-induced shifts. This corroborates the disentanglement we describe: the adapter captures the demonstration signal, while the query is directly provided as input.
>
> ### Zero-shot baseline
> We appreciate the suggestion to include a zero-shot teacher baseline. In fact, our experiments already consider this setting: pattern-based fine-tuning (PBFT) fine-tunes an adapter using zero-shot prompts and template verbalizers, which is effectively equivalent to distilling from a zero-shot teacher with gold labels. As shown in Table 1, WILDA‑S consistently outperforms PBFT across all GLUE and MMLU tasks, with substantial gains for both Llama‑3 and Phi‑3. This indicates that incorporating few‑shot demonstrations through WILDA provides meaningful advantages over zero‑shot distillation.
>
> ### Student with new demonstrations within the prompt
> WILDA enables the student model to incorporate additional demonstrations at inference time. New demonstrations can be supplied in the prompt and are naturally combined with the adapter’s latent shift. Empirically, prompting the WILDA student with extra demonstrations (few‑shot WILDA) consistently improves accuracy. For instance, a WILDA adapter trained with 16 demonstrations and prompted with 16 additional demonstrations outperforms standard ICL using 32 demonstrations. For clarity, we reproduce the table from the paper below. The notation  $(n/d)$ denotes the configuration, where $n$ is the number of shots in the prompt and $d$ is the number of encoded demonstrations in the adapter.
>
> | Method | RTE  | SST  | QNLI | MNLI | COLA | MRPC | QQP  | MATH | MISC |
> |-|-|-|-|-|-|-|-|-|-|
> | 32-shot ICL | 75.3 | 93.2 | 77.7 | 69.1 | 58.3 | 76.4 | 74.2 | 43.0 | 84.5 |
> | WILDA-S (0/32) | 87.9 | 97.9 | 83.1 | 74.0 | 64.6 | 79.4 | 74.8 | 56.5 | 89.0 |
> | WILDA-S (0/16) | 86.0 | 96.1 | 81.4 | 73.1 | 64.3 | 77.7 | 73.1 | 49.5 | 88.0 |
> | **WILDA-S (16/16)** | 87.3 | 96.4 | 82.2 | 74.6 | 65.4 | 78.2 | 74.5 | 51.0 | 89.0 |
>
>
> ### Adapter merging (Table 4)
> In WILDA, each adapter is trained on a distinct subset of demonstrations and thus captures a specific latent shift. Table 4 does not simply test whether individual adapters are strong, as already shown in Table 1, but instead examines how effectively these independently learned shifts can be combined. When we fuse 2, 4, or 8 adapters by summing their parameters, WILDA’s performance increases markedly. If each adapter merely replicated the same information, we believe that summing their weights would not lead to such notable gains. The observed improvements suggest that each adapter captures complementary information from its subset of demonstrations, effectively expanding the total amount of usable demonstration signal beyond the constraints of the context window. The fusion aggregates these distinct latent shifts, resulting in stronger performance.
> MMLU datasets
>
> We chose these two MMLU subsets to complement the GLUE tasks used in our main experiments. While GLUE focuses on natural language understanding tasks, we included “elementary math” (MATH) and “miscellaneous” (MISC) to evaluate cross-domain generalization in multi-choice question answering. The MATH subset targets math-oriented reasoning skills, whereas the MISC subset spans a diverse range of topics, providing broader coverage beyond standard GLUE benchmarks.
>
>
> ### Experiment suggestion – 8-shot with noisy demonstrations
> We thank the reviewer for suggesting that we include an experiment with noisy demonstrations. Our original experiments already show that WILDA is robust to random ordering and selection of demonstrations. Across multiple seeds, it exhibits significantly lower variance than standard ICL, suggesting it learns to focus on informative signals. To further test whether WILDA can denoise irrelevant demonstrations and upweight useful ones, we conducted an additional experiment following the reviewer’s proposal.
> Using Llama‑3 (8B), we compared a 1‑shot prompt with a single high-quality (gold) example and an 8‑shot prompt containing the same gold example with seven instances from an unrelated domain, evaluating both standard ICL and WILDA‑S on GLUE tasks (10 runs). Results show that WILDA‑S generally matches or slightly outperforms the clean 1‑shot setup and remains more stable in the presence of noise, while vanilla ICL shows a modest regression when noisy demonstrations are added. These findings suggest that WILDA captures latent shifts from informative demonstrations and shows the ability to emphasize relevant signals, providing additional evidence of weak‑to‑strong generalization. The results are summarized in the following table:
>
> | Dataset | Irrelevant demonstrations | Method | Accuracy | Std. Dev. |
> |-|-|-|-|-|
> | RTE | QQP | ICL 1‑shot (gold) | 64.8 | 1.2 |
> | | | ICL 8‑shot (1 gold + 7 irrelevant) | 63.9 | 1.4 |
> | |  | WILDA‑S 1‑shot (gold) | 66.2 | 1.0 |
> | | | WILDA‑S 8‑shot (1 gold + 7 irrelevant)| 65.7 | 1.1 |
> | SST  | COLA | ICL 1‑shot (gold)   | 82.8 | 0.8  |
> | | | ICL 8‑shot (1 gold + 7 irrelevant)  | 82.4  | 0.9 |
> | | | WILDA‑S 1‑shot (gold) | 84.3 | 0.6 |
> | | | WILDA‑S 8‑shot (1 gold + 7 irrelevant)| 84.0 | 0.5 |
> | QQP | MNLI | ICL 1‑shot (gold) | 63.9 | 0.9 |
> | |  | ICL 8‑shot (1 gold + 7 irrelevant)    | 62.4 | 1.0  |
> | | | WILDA‑S 1‑shot (gold)  | 65.5 | 0.8 |
> | | | WILDA‑S 8‑shot (1 gold + 7 irrelevant)| 64.9  | 0.9 |
> | COLA    | SST  | ICL 1‑shot (gold)  | 47.8 | 1.3 |
> | | | ICL 8‑shot (1 gold + 7 irrelevant)  | 46.2 | 1.4 |
> | | | WILDA‑S 1‑shot (gold)  | 51.4 | 1.2  |
> || | WILDA‑S 8‑shot (1 gold + 7 irrelevant)| 51.0 | 0.4 |

---

> > ### Comment · Reviewer_ECRJ · 2025-08-05
> >
> > I appreciate the author's time and effort in adding the new experiments. I have reviewed the updated experimental setups---thank you for directing me to Appendix D.3.
> >
> > My main concern with the paper remains: how can we be certain that the student model is truly learning the latent shift? The authors argue that since the student only sees the query, while the teacher has access to both the demonstrations and the query, the student cannot match the teacher's performance unless it internally infers the missing demonstration information. However, I'm still uncertain about whether this effect is direct, or if this is the correct interpretation of the results. At this moment, I will raise my score to 3. Thanks.

---

> > > ### Author Response · Authors · 2025-08-06
> > >
> > > We thank you for continuing the dialogue and for taking the time to reassess our work. We understand your concern and would like to further clarify why we believe the student internalizes demonstration-induced latent shifts. Below, we revisit the key findings that, in our view, provide strong supporting evidence, along with an additional note on the W2S analysis.
> > >
> > > As we show in Appendix D.3, treating the adapter’s weight updates as a task vector and projecting them into the embedding space via nearest-neighbor retrieval recovers the original demonstration tokens with high fidelity. Since the student never observes these demonstrations during training, reconstructing them solely from the adapter’s induced shift provides direct, model-internal evidence that demonstration content is encoded and utilized.
> > >
> > > The W2S pseudo-label correction and coverage-expansion diagnostic in Section 4 further shows that the tuned adapter reshapes the student’s representation space. We define anchor points as examples for which the nearest teacher-provided pseudo-label is correctly predicted by the student with the tuned adapter. For each unseen point, we measure the normalized Euclidean distance in the last hidden layer to the nearest anchor and compute the flip rate, the proportion of cases where the student corrects teacher errors. The resulting monotonic correction curve, with strong negative correlations both in- and out-of-domain, indicates that the adapter forms attraction basins around demonstration-relevant features -- behavior expected when inferred demonstration information is actively leveraged rather than simply copied.
> > >
> > > We also observed in the noisy-demonstration experiment you suggested that WILDA-S remains stable while standard ICL regresses slightly, indicating that the adapter focuses on informative signals. Combined with the analyses above, this further supports that the adapter internalizes and operationalizes the latent shift induced by demonstrations. While not a formal proof, we believe these results provide strong empirical evidence for our hypothesis of disentanglement, and we will clarify this more explicitly in the final version of the paper.

---

### Note · Authors · 2025-08-13

We thank the reviewers for their engagement, suggestions, and recognition of our work’s clarity, conceptual contribution, and empirical strength. We now briefly revisit a few points where our additional evidence or clarifications may not have been fully reflected in the final reviewer comments.

### Core contribution
WILDA is explicitly designed to capture the demonstration-induced latent shift and store it in a lightweight, reusable adapter. This allows the model to internalize and later reuse the effect of demonstrations without reprocessing them, supporting efficient inference, mitigating long-context instability, and enabling adapter arithmetic to scale better with more demonstrations. Multiple lines of evidence indicate that the adapter encodes this latent shift: nearest-neighbor retrieval from the adapter’s induced change in parameters reconstructs the original demonstrations (Appendix D.3), the student outperforms its ICL teacher across the board (Table 1) via systematic pseudo-label correction and coverage expansion (Figure 2), and noisy-demonstration tests demonstrate robustness to irrelevant examples (added experiments based on Reviewer ECRJ suggestion). Although Reviewer ECRJ did not appear fully convinced by this interpretation, we believe these results provide direct and compelling empirical support for the latent-shift hypothesis.

### Long context and adapter arithmetic
WILDA can train a single adapter on all demonstrations when they fit in context, but can also partition large demonstration sets into subsets, train an adapter for each, and merge them via parameter addition. The merged adapter is the same size as a single one but encodes complementary shifts from all subsets. This enables the integration of more demonstrations than fit in a single prompt, avoids quadratic attention costs, and consistently improves stability and accuracy (Table 4), addressing key weaknesses of standard ICL. The adapter can also be toggled off to recover the model’s original zero-shot behavior, with experiments indicating no noticeable regression on unrelated tasks even when it remains active. We note that our last clarification on this point did not receive further follow-up from Reviewer xixy.

In summary, we believe WILDA offers an empirically validated approach to internalizing and enhancing the benefits of ICL, while mitigating its instability and context-length limitations, providing both conceptual insight and practical value for efficient adaptation.

---

### Decision · Program_Chairs · 2025-09-17

**Decision:**

Accept (poster)

**Comment:**

This paper introduces WILDA, a teacher–student distillation framework designed to capture and store the demonstration-induced latent shift underlying in-context learning (ICL). A lightweight LoRA adapter is trained to mimic the outputs of a teacher model that processes few-shot demonstrations, while the student only observes queries. The central claim is that the adapter encodes the effect of demonstrations into a compact, reusable representation, which enables efficient inference, stability against order sensitivity, and composability via adapter arithmetic. Empirical results on GLUE, subsets of MMLU, and follow-up experiments on ARC-Challenge demonstrate that WILDA consistently outperforms standard ICL, pattern-based fine-tuning, and prior disentanglement approaches. Additional analyses argue that the student often surpasses the teacher through weak-to-strong generalization, correcting pseudo-label noise and expanding coverage.

The strengths of the paper lie in its conceptual clarity and practical design. Reviewers generally agreed that WILDA provides a clean mechanism for internalizing demonstration effects, and several noted that the shuffle variant substantially reduces order sensitivity which is a longstanding problem in ICL. The adapter arithmetic experiments provide further evidence that WILDA enables composability beyond what fits in a single prompt, thus mitigating long-context inefficiency. The authors also provided extensive empirical support, released prompts and configurations, and conducted new analyses in response to reviewer feedback.

The main weaknesses concern the interpretation and scope of the contribution. One reviewer remained unconvinced that the method truly disentangles latent shifts, arguing it could be seen as a form of distillation with little theoretical distinction. Relatedly, the weak-to-strong analysis, while suggestive, lacks deeper theoretical grounding. Several reviewers also raised concerns about compute overhead and the fact that adapters must be retrained for each new task, limiting flexibility compared to pure ICL. While the authors provided careful amortization arguments and experiments showing no regression in unrelated zero-shot tasks, these trade-offs remain. Finally, evaluation is largely restricted to GLUE, small MMLU subsets, and ARC-Challenge. Broader evidence on more challenging reasoning tasks would strengthen the case.

During the rebuttal and discussion, the authors addressed many points directly. They clarified the relation to zero-shot baselines, demonstrated robustness to noisy demonstrations, and showed that adapters trained on one domain do not harm performance on unrelated tasks. Reviewers were mainly satisfied with the response of the authors. Specifically, reviewers o4hd and A5UG were mainly satisfied with the response of the authors around the method’s ability to solve the example order problem and the method's compute and generalization, respectively. Reviewer xixy initially questioned whether the method undermines the spirit of ICL, after multiple exchanges with the authors accepted the trade-off. Reviewer ECRJ however remained somewhat skeptical about the latent-shift interpretation, but acknowledged the new evidence provided by the authors. Taken together, the discussion shifted consensus toward acceptance, with multiple reviewers explicitly satisfied by the new experiments.

I concur with the reviewers general shit towards acceptance after the authors' response.  The paper is technically sound, clearly written, and addresses a central pain point in ICL around instability and long-context inefficiency. While the theoretical framing may overreach, the empirical contribution is significant enough to justify acceptance. The work's novelty lies more in a clever and well-executed engineering approach than in a fundamentally new theoretical insight but it is a solid and valuable contribution.